



# CALIPSO observations of the dependence of homo- and heterogeneous ice nucleation in cirrus clouds on latitude, season and surface condition

David L. Mitchell[1], Anne Garnier[2], Melody Avery[3], and Ehsan Erfani[4]

[1]Desert Research Institute, Reno, 89512-1095, USA

[2]Science Systems and Applications, Inc., Hampton, Virginia, USA

[3]NASA Langley Research Center, Hampton, Virginia, USA

[4]George Mason University, Fairfax, Virginia, USA

*Correspondence to*: David Mitchell (david.mitchell@dri.edu)

**Abstract.**  There are two fundamental mechanisms through which cirrus clouds form; homo- and heterogeneous ice nucleation (henceforth hom and het).  The relative contribution of each mechanism to ice crystal production often determines the microphysical and radiative properties of a cirrus cloud.  A new satellite remote sensing method is described in this study to estimate cirrus cloud ice particle number concentration and the relative contribution of

hom and het to cirrus cloud formation as a function of altitude, latitude, season and surface type (e.g. land vs. ocean).  This method uses co-located observations from the Infrared Imaging Radiometer (IIR) and from the CALIOP (*Cloud and Aerosol Lidar with Orthogonal Polarization*) lidar aboard the CALIPSO (*Cloud-Aerosol Lidar and Infrared Pathfinder Satellite Observation)* polar orbiting satellite, employing IIR channels at 10.6 µm and 12.05 µm.  The method is applied here to single-layered clouds of visible optical depth between about 0.3 and 3.  Two

years of Version 3 data have been analyzed for the years 2008 and 2013, with each season characterized in terms of 532 nm cirrus cloud centroid altitude and temperature, the cirrus cloud ice particle number concentration, effective diameter, layer-average ice water content and visible optical depth.  Using a conservative criterion for hom cirrus, on average, the sampled cirrus clouds formed through hom occur about 43% of the time in the Arctic and 50% of the time in the Antarctic, and during winter at mid-latitudes in the Northern Hemisphere, hom cirrus occur 37% of the

time.  Elsewhere (and during other seasons in the Northern Hemisphere mid-latitudes), this hom cirrus fraction is lower.  Processes that could potentially explain these observations are discussed, as well as the potential relevancy of these results to ice nucleation studies, climate modeling and jet-stream dynamics.





## 1  Introduction

The Fifth Assessment Report (AR5) of the Intergovernmental Panel on Climate Change (IPCC 2013) states that "Climate models now include more cloud and aerosol processes, and their interactions, than at the time of AR4, but there remains low confidence in the representability and quantification of these processes in models." The largest
single cause of uncertainty in anthropogenic radiative forcing is from the indirect effect of aerosols on clouds, and the dependence of the microphysical properties of cirrus clouds on natural and anthropogenic aerosols is perhaps the least understood of these indirect effects.

Cirrus clouds are pure ice clouds that form at temperatures less than 235 K (where liquid water cannot exist). They are different than liquid water clouds in that they can be formed in two different ways; through homogeneous and
heterogeneous ice nucleation (henceforth hom and het, respectively). Het requires an insoluble aerosol particle like mineral dust to serve as an ice nuclei (IN) that initiates the ice phase through direct deposition of water vapor onto the IN, or by the IN initiating freezing from within an aqueous medium (such as a cloud droplet). Conversely, hom cirrus do not require a particle, with ice forming through the freezing of pure solution droplets such as in haze or cloud droplets (Koop, 2000). Without understanding the relative contributions of these two mechanisms to cirrus
cloud formation, and how this varies with temperature, latitude, season and surface type, the aerosol indirect effect for cirrus clouds will remain obscure. Moreover, these two formation mechanisms can lead to major differences in cirrus cloud microphysical properties that determine their radiative properties, with the ice particle number concentration N generally being much higher in hom cirrus (e.g. Barahona and Nenes, 2009). For purposes of climate impact, it may be useful to subdivide cirrus into two categories: hom cirrus and het cirrus, where hom
dominates the microphysical properties in hom cirrus, and conversely for het cirrus. The factors that determine whether hom or het dominate may depend on the amplitude of atmospheric wave activity (which affect the air parcel cooling rate; e.g. Haag et al., 2003), the IN concentration at cirrus levels, and whether nucleation occurs in the presence of pre-existing ice (e.g. Shi et al., 2015), and these may be a function of surface topography (e.g. mountainous terrain vs. ocean or plains), latitude and season.

Krämer et al. (2016a) evaluated cirrus cloud in situ data from many field campaigns using a detailed cirrus cloud model to infer the microphysical processes responsible for the observations. They found that hom dominated in cirrus clouds that form in fast updrafts produced by atmospheric waves (e.g. leewave cirrus). These cirrus may be associated with mountainous terrain. Otherwise the cirrus tended to result from het. The "liquid origin cirrus" described in Krämer et al. (2016a) and Luebke et al. (2016) for ice clouds having T < 235 K, are formed primarily
by the heterogeneous freezing of liquid droplets at temperatures T > -38°C (235 K), with possible contributions from the homogeneous freezing of cloud droplets that occurs only at ~ 235 K, and from hom (T < 235 K) if the updraft is





fast enough. Ice produced at T > 235 K is lifted from these lower levels into the "cirrus zone" where T < 235 K. Liquid origin cirrus include anvil cirrus and some frontal cirrus fed by ascending "conveyer belts" of moisture.

While these and other studies have advanced our understanding of how cirrus clouds form and thus their microphysical and radiative properties, they draw from a limited number of field campaigns that may not represent

all the conditions in which cirrus clouds form, and may not provide a reliable understanding of the geographic and seasonal distribution of het and hom cirrus. For example, there have been few cirrus cloud field campaigns in the Polar Regions and in the Southern Hemisphere where IN concentrations are predicted to be much lower relative to the Northern Hemisphere. Without a global monitoring system capable of inferring cirrus cloud formation processes, global climate model (GCM) predictions of cirrus cloud radiative forcing may not be realistic, especially

if the predicted geographic and seasonal dependence of hom and het cirrus clouds is flawed.

This study describes a new approach for estimating cloud layer N, the ice particle size distribution (PSD) effective diameter $D_e$, and the layer-average ice water content (IWC) in selected semi-transparent cirrus clouds using co-located observations from the 10.6 μm and 12.05 μm channels on the Imaging Infrared Radiometer (IIR) aboard the CALIPSO (*Cloud-Aerosol Lidar and Infrared Pathfinder Satellite Observation)* polar orbiting satellite, augmented

by the cloud layer 532 nm centroïd altitude, cloud geometric thickness and extinction profile from CALIOP (*Cloud and Aerosol Lidar with Orthogonal Polarization*) and by interpolated temperatures from a GMAO reanalysis. CALIOP and IIR are assembled in a near-nadir looking configuration. The cross-track swath of IIR is by design centered on the CALIOP track where observations from the two instruments are perfectly temporally collocated. The spatial co-location is nearly perfect, as CALIOP samples the same cloud as the 1-km IIR pixel, but with three

laser beam penetrations per km having a horizontal footprint of 90 m. While the IIR retrieves layer-average cloud properties, CALIOP retrieves vertical profiles within this layer, providing additional information to guide the retrievals and enhance the interpretation. These retrieved cloud properties are compared with corresponding cloud properties measured *in situ* during several field campaigns conducted throughout the world. Some initial IIR retrieval results are reported for 2008 and 2013 during each season for all latitudes. These results are used to

formulate working hypotheses that explain some mechanisms for the seasonal dependency of the global distribution of hom and het cirrus.

Section 2 describes the rationale for developing retrieval results, Sect. 3 describes the retrieval physics and methodology, while comparisons between retrieved and measured cloud properties, as well as retrieval uncertainties, are discussed in Sect. 4. Retrieval results and discussions thereof are given in Sect. 5, and other studies are related

to this study in Sect. 6. Conclusions are presented in Sect. 7.



## 2 Discrimination between het and hom

This section describes the method for discriminating between hom and het cirrus clouds and the rationale for using this method. As shown in Barahona and Nenes (2008), for a given value of the water vapor deposition coefficient, ice crystal production rates by hom are most sensitive to the cloud updraft (i.e. the cooling rate), then temperature

and the near-cloud aerosol concentration (comparable sensitivities for both), and they are least sensitive to the mean aerosol particle size. The cooling rate is a key factor determining the relative humidity with respect to ice (RHi) in a rising air parcel, and hom requires a RHi > 145%, depending on temperature. Since N due to hom results from the freezing of haze and cloud solution droplets (the larger sizes in their PSDs), and their number concentrations are typically > 200 L⁻¹, (reaching concentrations of 10,000 L⁻¹ and higher), hom is generally associated with relatively

high N. Under very cold and unique conditions of weak and relatively short-lived updrafts (e.g. low-amplitude gravity waves), N resulting from hom may be < 100 L⁻¹ (Sprichtinger and Krämer, 2013; Krämer et al., 2016a). On the other hand, ice crystal production rates from het depend mostly on the IN concentration and composition (Pruppacher and Klett, 1997, Ch. 9), with IN comprising a very small subset of the total aerosol population. Due to this, N resulting from het tends to be less than ~ 200 L⁻¹ (Barahona and Nenes, 2009; Jensen et al., 2012a,b; Cziczo

et al., 2013), although higher concentrations are possible under atypical conditions. Moreover, the RHi required for het typically ranges from 100% to 140%, depending on IN composition (e.g. Kärcher and Lohmann, 2003; Kärcher et al, 2007).

In theory, one could use either N or RHi to discriminate between hom and het. But in practice, for satellite remote sensing to discriminate between hom and het conditions among cirrus clouds, the best distinguishing feature to

exploit appears to be the generally observed differences in N. In Barahona and Nenes (2009, Fig. 4), competition effects between het and hom are simulated in a parcel model using a broad spectrum of conditions (affecting nucleation) found in nature. They find that het generally accounts for N < 200 L⁻¹ and that either hom or het can account for N between 200 L⁻¹ and 500 L⁻¹, while hom generally accounts for N > 500 L⁻¹. Hence, in this study we use N > 500 L⁻¹ as a conservative threshold for cirrus dominated by hom. Although RHi differs between hom and

het at the time of nucleation, RHi tends to rapidly decrease after the onset of hom due to vigorous competition for water vapor among the ice crystals occurring at relatively high N. This may produce lower RHi under hom conditions relative to het conditions (e.g. Jensen et al., 2013), rendering RHi an uncertain means of discriminating between hom and het. In addition to N, topography (related to cooling rates) can also be used to infer nucleation mechanisms based on the attributes noted above. For example, mountainous terrain will induce relatively high

amplitude atmospheric waves at cirrus cloud levels (Jiang et al., 2002; 2004; Wu and Jiang, 2002; Hoffmann et al., 2016), resulting in higher cloud updrafts. Thus mountainous terrain in association with relatively high N should be a strong indicator for hom cirrus. Hom cirrus may also be more likely to occur in regions having relatively low IN concentrations, such as the Polar Regions. To the best of our knowledge, a satellite remote sensing method sensitive





to N (which is dominated by the smallest ice crystals) has not been developed, and a critical objective of this study was to develop such a method.

### 3    Developing a satellite remote sensing method sensitive to N

It is widely recognized that the ratio of absorption optical depth from ice clouds, β, based on wavelengths at 12 μm

and 11 μm (or similar wavelengths), is rich in cloud microphysical information (Inoue, 1985; Parol et al., 1991; Cooper et al., 2003; Dubuisson et al., 2008; Heidinger and Pavolonis, 2009; Pavolonis, 2010; Mitchell et al., 2010; Cooper and Garrett, 2010; Garnier et al., 2012; Mitchell and d'Entremont, 2012; Garnier et al., 2013). These studies have used a retrieved β to estimate the effective diameter $D_e$, the ice water path (IWP), the mass-weighted ice fall speed ($V_m$), the average fraction of liquid water in a cloud field and the relative concentration of small ice crystals in

ice PSDs. However, the main reason for the emissivity differences in satellite remote sensing channels centered on these two wavelengths was not understood until after the development of the modified anomalous diffraction approximation (MADA) that, to a first approximation, allowed various scattering/absorption processes to be isolated and evaluated independently (Mitchell, 2000; Mitchell et al., 2001; Mitchell, 2002; Mitchell et al., 2006). For wavelengths between 2.7 and 100 μm, the most critical process parameterized was wave resonance, also referred to

as photon tunneling (e.g. Nussenzveig, 1977; Guimaraes and Nussenzveig, 1992; Nussenzveig, 2002). It was this process that was found to be primarily responsible for the cloud emissivity difference between these wavelengths (12 μm and 11 μm) in ice clouds, as described in Mitchell et al. (2010). The greatest tunneling contribution to absorption occurs when the ice particle size and wavelength are comparable and the real refractive index $m_r$ is relatively high (Mitchell, 2000). Since $m_r$ at 12 μm is high relative to $m_r$ for the 11 μm wavelength, β is sensitive to

the tunneling process and the relative concentration of small (D < 60 μm) ice crystals in cirrus clouds.

It was originally thought that β resulted from differences in the imaginary index of ice, $m_i$, at two wavelengths (λ) near 11 μm and 12 μm, but it is actually due to differences in the real index of refraction, $m_r$. At these λ, $m_i$ is sufficiently large so that most ice particles in the PSD experience area-dependent absorption (i.e. no radiation passes through the particle), and the absorption efficiency $Q_{abs}$ for a given ice particle will be ~ 1.0 for both λ when $Q_{abs}$ is

based only on $m_i$ (i.e. the $Q_{abs}$ predicted by Beer's law absorption or anomalous diffraction theory). The observed difference between $Q_{abs}$(12 μm) and $Q_{abs}$(11 μm) is due to differences in the photon tunneling contribution to absorption that primarily depends on $m_r$ (Mitchell, 2000). That is, $m_r$ is substantial when λ = 12 μm but is relatively low when λ = 11 μm, producing a substantial difference between $Q_{abs}$(12 μm) and $Q_{abs}$(11 μm). Figure 1 shows the size dependence of the tunneling contribution for hexagonal columns at 12 μm. It is evident that this contribution

becomes important only for the smallest ice crystal sizes where wavelength and crystal size are comparable (i.e. maximum dimension D < 50 μm), making β well suited for detecting recently nucleated (small) ice crystals that primarily determine N (Krämer et al., 2009).





In this study, we use CALIPSO IIR channels at 10.6 μm and 12.05 μm and define β as

$$\beta = \tau_{abs}(12.05\ \mu m)/\tau_{abs}(10.6\ \mu m) \tag{1}$$

where $\tau_{abs}$ is the retrieved absorption optical depth for a given λ retrieved from the effective emissivity. However, what is retrieved is not exactly β, but a β that also includes the effects of scattering, defined as the effective β or $\beta_{eff}$.

$\beta_{eff}$ is described analytically in Parol et al. (1991). For a given cirrus cloud, retrieved $\tau_{abs}(10.6\ \mu m)$ may be slightly less than $\tau_{abs}(11\ \mu m)$ since $m_i$ at 10.6 μm is less than $m_i$ at 11 μm, meaning that some Beer's law type absorption may contribute to emissivity differences between the 10.6 μm and 12 μm channels when PSD are sufficiently narrow. This acts to extend the dynamic range of retrievals relative to the 11 μm-12μm channel combination (e.g. the limiting maximum $D_e$ retrieved will be greater using 10.6 and 12 μm relative to 11 and 12 μm). The

methodology for retrieving CALIPSO IIR effective emissivity and $\beta_{eff}$ from co-located CALIOP observations and IIR radiances is described in Garnier et al. (2012, 2013). IIR retrievals are from the CALIPSO IIR Version 3 Level 2 track product (Vaughan et al., 2015). In this product, the scene typing is built from the CALIOP Version 3 5-km cloud and aerosol layer products. It has been refined for this study to account for additional dense clouds in the planetary boundary layer reported in the CALIOP Version 3 333-m layer product. IIR retrievals are further

corrected to reduce possible biases, as described in Sect. 3.2. Version 3 CALIOP cloud extinction coefficient profiles are used for some of the corrections. These improvements will be implemented in the next version 4 of the IIR products.

### 3.1   Relating $\beta_{eff}$ to N/IWC and $D_e$ based on aircraft PSD measurements

Using aircraft data from the DOE ARM supported Small Particles in Cirrus (SPartICus) field campaign in the

central United States and the NASA supported Tropical Composition, Cloud and Climate Coupling (TC4) field campaign near Costa Rica, $\beta_{eff}$ was related to cirrus cloud microphysical properties. Regarding SPARTICUS, the data set described in Mishra et al. (2014) was used, and the TC4 data is described in Mitchell et al. (2011). Details regarding field measurements, the flights analyzed and the microphysical processing are described in these articles. The PSDs were measured by the 2D-S probe (Lawson et al., 2006) where ice particle concentrations were measured

down to 10 μm (5–15 μm size bin) and up to 1280 μm in ice particle length (when using "all-in" data processing criteria). The data in the smallest size bin (5–15 μm) has greater uncertainty. Indeed, Jensen et al. (2013) showed that the largest uncertainty in depth of field for this size bin results in an overestimation of number concentration for particles in this smallest size bin. $\beta_{eff}$ was calculated from these PSDs using the method described in Parol et al. (1991) and Mitchell et al. (2010). This method was tested in Garnier et al. (2013, Fig. 1b) where $\beta_{eff}$ calculated from

a radiative transfer model (FASDOM; Dubuisson et al., 2005) was compared with $\beta_{eff}$ calculated analytically via Parol et al. (1991), with good agreement found between these two methods. More specifically, to calculate $\beta_{eff}$ from



PSD in this study, the PSD absorption efficiency $\overline{Q}_{abs}$ is given as $\overline{Q}_{abs} = \beta_{abs} / A_{PSD}$, where $\beta_{abs}$ is the PSD absorption coefficient (determined by MADA from measured PSD) and $A_{PSD}$ is the measured PSD projected area. The PSD effective diameter was determined from the measured PSD as described in Mishra et al. (2014), but in essence is given as $D_e = (3/2)\ IWC/(\rho_i\ A_{PSD})$, where $\rho_i$ is the density of bulk ice (0.917 g cm$^{-3}$). The PSD extinction

efficiency $\overline{Q}_{ext}$ was determined in a manner analogous to $\overline{Q}_{abs}$. The single scattering albedo $\omega_o$ was calculated as $\omega_o = 1 - \overline{Q}_{abs}/\ \overline{Q}_{ext}$ and the PSD asymmetry parameter g was obtained from $D_e$ using the parameterization of Yang et al. (2005). Knowing $\overline{Q}_{abs}$, $\omega_o$ and g, $\beta_{eff}$ was calculated from the PSD as:

$$\beta_{eff} = Q_{abs,eff}(12\ \mu m)/Q_{abs,eff}(11\ \mu m)\ , \tag{2}$$

$$Q_{abs,eff} = Q_{abs}\ (1 - \omega_o\ g)\ /\ (1 - \omega_o). \tag{3}$$

Note that $\beta$ (i.e. $\beta_{eff}$ without scattering effects) is also equal to $\overline{Q}_{abs}(12\ \mu m)/\ \overline{Q}_{abs}(10.6\ \mu m)$.

In order to retrieve N, a relationship between N/IWC and $\beta_{eff}$ is useful. Figure 2 shows measurements of N/IWC from SPARTICUS flights over the central United States (blue) where some of the cirrus sampled (i.e. the "ridge crest cirrus"; see Muhlbauer et al., 2015) had high N (500-2200 L$^{-1}$) for T < -60°C. Also shown are N/IWC measurements from the TC4 field campaign for maritime "fresh" anvil cirrus (during active deep convection where

the anvil is linked to the convective column) and for TC4 aged anvil cirrus (anvils detached from convective column). Figure 2 relates $\beta_{eff}$ to the N/IWC ratio, where $\beta_{eff}$ was calculated from the same PSD measurements used to calculate N and IWC, based on the MADA method. The PSD measurements include size-resolved estimates of ice particle mass concentration based on Baker and Lawson (2006), size-resolved measurements of ice projected area concentration, and the size resolved number concentration. This PSD information is the input for the MADA

method that yields the coefficients of absorption and extinction. The tunneling efficiency $T_e$ used in MADA was estimated from Table 1 in Mitchell et al. (2006), where for 1 μm < D < 30 μm, droxtals and hexagonal columns are assumed and $T_e = 0.90$; for 30 μm < D < 100 μm, budding bullet rosettes and hexagonal columns are assumed and $T_e = 0.50$; for D > 100 μm bullet rosettes and aggregates are assumed and $T_e = 0.15$. This shape-dependence on ice particle size was guided by the ice particle size-shape observations reported in Lawson et al. (2006) and Baker and

Lawson (2006). These ice particle shape assumptions affect only $T_e$, and the cloud optical properties are primarily determined through the PSD measurements noted above (i.e. not the value of $T_e$). Due to $\beta_{eff}$'s sensitivity to tunneling and small ice crystals, a tight and useful relationship is found between N/IWC and $\beta_{eff}$ for N/IWC > ~ $10^7 g^{-1}$. As far as we know, this relationship was not known previously. For $\beta_{eff}$ < 1.035, $\beta_{eff}$ is not sensitive to N/IWC and N/IWC cannot be estimated from $\beta_{eff}$. For purposes of discriminating hom cirrus from het cirrus, the $\beta_{eff}$

vs. N/IWC relationship appears ideal since the hom-het transition generally occurs in the region where $\beta_{eff}$ is



sensitive to changes in N/IWC (i.e. when N/IWC > $5 \times 10^7$ g$^{-1}$, based on in situ PSD and the hom-het transition region described in Sect. 2).

Using this same in situ data and methodology, $\beta_{eff}$ has also been related to $1/D_e$ as shown in Fig. 3. $D_e$ is defined as (3/2) IWC/($\rho_i$ A$_{PSD}$) where $\rho_i$ is the density of bulk ice (Mitchell, 2002). Accordingly, $D_e$ was calculated from the

measured PSD (see Mishra et al., 2014). The relationship is only useful for $D_e$ < 90 µm since $\beta_{eff}$ is only sensitive to the smaller ice particles. $\beta_{eff}$ is a measure of the relative concentration of small ice crystals in a PSD (Mitchell et al., 2010), and A$_{PSD}$ and $\beta_{eff}$ (PSD integrated quantities) may be associated with a substantial portion of larger ice particles (D > 50 µm) before $\beta_{eff}$ loses sensitivity to changes in $D_e$.

Figure 4 compares $\beta_{eff}$ calculated from 2D-S probe measurements of TC4 and SPARTICUS PSDs based on MODIS

channels at 11 and 12 µm with the mean value of MODIS retrievals of $\beta_{eff}$ from TC4 cirrus clouds (i.e. during the TC4 campaign) using the methodology and results described in Mitchell and d'Entremont (2012; see their Fig. 5) whereby the cloud temperature is also retrieved. The high in situ derived $\beta_{eff}$ values for TC4 near -70°C are characterized by lower IWCs, and thermal emissions from these cloud levels may have been below the detection threshold of MODIS. In general there is good agreement between the TC4 measurement derived values and the

retrieved mean values of $\beta_{eff}$ (dashed line) during TC4. $\beta_{eff}$ derived from SPARTICUS data is also included to better illustrate the temperature dependence of $\beta_{eff}$ at warmer and colder temperatures. Note that the range of $\beta_{eff}$ has changed due to the MODIS channels used here.

It is noteworthy that retrieved $\beta_{eff}$ is often quasi-constant with T for TC4 cirrus when T < 235 K, as shown in Fig. 5 of Mitchell and d'Entremont (2012). This "flat" $\beta_{eff}$ behavior appears to result from the variation in ice crystal shape

with D across the PSD. That is, tunneling contributions depend on the distribution of ice particle shape with ice particle size D across the PSD. If the shape is not varied, then this flat behavior is not obtained, as shown in Fig. 12 of Mitchell and d'Entremont (2012) for the Yang et al. (2005) scheme where shape is held constant. This flat $\beta_{eff}$ signature provides some constraint for determining reasonable assumptions for the variation of ice crystal shape with D.

These range restrictions (N/IWC > ~ $10^7$ g$^{-1}$ & $D_e$ < 90 µm) are usually compatible with cirrus clouds (T < 235 K) since PSDs tend to be narrower at these temperatures, containing relatively small ice particles (e.g. Mishra et al., 2014). When calculating N/IWC from $\beta_{eff}$, if the retrieved $\beta_{eff}$ is less than 1.035, then $\beta_{eff}$ is set to a value of 1.035 and N/IWC corresponds to this value via the regression curve (N/IWC = $5.6 \times 10^7$ g$^{-1}$). Similarly, when calculating $D_e$ from $\beta_{eff}$, if the retrieved $\beta_{eff}$ is less than 1.0, then $\beta_{eff}$ is set equal to 1.0 and $D_e$ is calculated from that value ($D_e \approx$

122 µm). As shown in Table 1 for the year 2013, this practice affected about 14% and 20% of the N/IWC retrievals over ocean and land, respectively, and affected about 6% and 11% of the $D_e$ retrievals over ocean and land, respectively. This produces a line of constant valued N/IWC and $D_e$ retrievals when $\beta_{eff}$ drops below these threshold

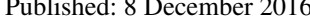



values (evident in Figs. 7, 8, 22 and 23). To better estimate the mean and median values for N/IWC and $D_e$, these

limiting values are taken into account when calculating these quantities.

### 3.2 The retrieval equations

These IIR layer-average N/IWC and $D_e$ relationships are combined to retrieve the ice particle number concentration

5    N, as shown in the equation:

$$N = \frac{\rho_i}{3} \times \frac{\left[2/\overline{Q}_{abs}(12\mu m)\right]\tau_{abs}(12\mu m)}{\Delta z_{eq}} \times D_e \times \left(\frac{N}{IWC}\right) \tag{4}$$

where $\rho_i$ is the bulk density of ice (0.917 g cm$^{-3}$), $\tau_{abs}$(12 µm) is the retrieved absorption optical depth from the IIR

12 µm channel, and $\Delta z_{eq}$ is the effective cloud thickness. The quantities $D_e$ and N/IWC are retrieved from $\beta_{eff}$, using

the regression curves described above, in Figs. 4 and 5. The quantity 2/ $\overline{Q}_{abs}$(12 µm) is obtained from $\beta_{eff}$ via the

10   regression equation described in Fig. 5, where 2 is the value of $\overline{Q}_{ext}$ for ice PSDs in the visible spectrum. When $\beta_{eff}$

> 1.485, then 2/ $\overline{Q}_{abs}$(12 µm) is set to 1.57. This quantity converts $\tau_{abs}$(12 µm) to an equivalent visible extinction

optical depth (OD) . The right hand side of (4), excepting the N/IWC term, is an expression for the layer-mean

IWC, which was derived from the familiar equation:

$$IWC = \frac{\rho_i}{3} \times \alpha_{ext} \times D_e \tag{5}$$

15   where in this circumstance $\alpha_{ext}$ is the effective layer-mean visible extinction coefficient, which was derived from the

CALIPSO IIR as:

$$\alpha_{ext} = \frac{\left(2/\overline{Q}_{abs}(12\mu m)\right)}{\Delta z_{eq}} \times \tau_{abs}(12\mu m). \tag{6}$$

For each IIR channel, $\tau_{abs}$ is derived from the effective emissivity, $\varepsilon$, as

$$\tau_{abs} = -\ln(1-\varepsilon) \tag{7}$$

20   with

$$\varepsilon = \frac{R_m - R_{BG}}{R_{BB} - R_{BG}} \tag{8}$$





where $R_m$ is the measured calibrated radiance, $R_{BB}$ is the opaque (i.e. blackbody) cloud radiance evaluated at cloud temperature $T_c$, and $R_{BG}$ is the background radiance that would be observed in the absence of the studied cloud, as described in Garnier et al. (2012).

The brightness temperature $T_{BG}$ associated to the background radiance $R_{BG}$ is derived from the FASt RADiative
transfer model (Dubuisson et al., 2005) fed by atmospheric profiles and skin temperatures along with pre-defined surface emissivities (Garnier et al., 2012). For this study, remaining biases at 12.05 μm and 10.6 μm are corrected using monthly maps of mean differences between observed and computed brightness temperatures (called BTDoc) in clear sky conditions. The corrections are applied over ocean and over land with a resolution of 2 degrees in latitude and 4 degrees in longitude, by separating daytime and nighttime data. After correction, BTDoc is equal to
zero on average for both channels.

In version 3 IIR products, $R_{BB}$ (and the associated blackbody temperature $T_{BB}$) is derived from the cloud temperature, $T_c$, evaluated at the centroid altitude of the CALIOP 532 nm attenuated backscatter profile using GMAO reanalysis temperature (Garnier et al, 2012). In this study, a correction for residual biases is determined using CALIOP extinction profiles in the cloud layer as described in Sect. 3 of Garnier et al. (2015). The CALIOP
lidar 532 nm extinction profile in the cloud is used to determine a weighting profile that is used to compute $R_{BB}$ as the weighted averaged blackbody radiance. The lidar vertical resolution is 60-m, and cloud emissivity is weighted in a similar way with the 532 nm extinction profile. The weight of each 60-m bin is its emissivity at 12.05 μm attenuated by the overlying optical depth, normalized to the cloud emissivity.

The effective cloud thickness $\Delta z_{eq}$ accounts for the fact that the IIR instrument does not sense equally all of the
cloud profile that contributes to thermal emission. Following the same approach as described above, the layer absorption coefficient $\alpha_{abs}(12\ \mu m)$ for the IIR 12 μm channel is computed from the weighted averaged absorption coefficient profile. This yields $\alpha_{abs}(12\ \mu m) > \alpha_{abs,mean}(12\ \mu m)$ , where $\alpha_{abs,mean}(12\ \mu m)$ is the mean absorption coefficient, that is, the ratio of $\tau_{abs}(12\ \mu m)$ to the CALIOP layer geometric thickness $\Delta z$. Thus, an equivalent effective thickness is defined as $\Delta z_{eq}$ where $\alpha_{abs}(12\ \mu m) = \tau_{abs}(12\ \mu m)/ \Delta z_{eq}$ , or alternatively, $\Delta z_{eq} = \Delta z\ (\alpha_{abs,mean}(12$
$\mu m)\ /\ \alpha_{abs}(12\ \mu m))$. In practice, $\Delta z_{eq}$ is found equal to 30% to 90 % of $\Delta z$.

To summarize, the retrieval of $\tau_{abs}(12\ \mu m)$ and $\tau_{abs}(10.6\ \mu m)$ combined with the CALIOP extinction profile provides $N/IWC$, $D_e$, $\alpha_{ext}$, and therefore layer-average IWC and finally N. It also provides the ice water path (IWP) as

$$IWP = \frac{\rho_i}{3} \times \left(2/\overline{Q}_{abs}(12\mu m)\right) \times \tau_{abs}(12\mu m) \times D_e. \tag{9}$$



Since the IWP can also be expressed as $(\rho_i/3)\, D_e\, OD$ where OD is visible optical depth, and since upper limits for $D_e$ and OD are 122 µm and 3.0, respectively, the IWP upper limit is 112 g m$^{-2}$. Perhaps the most unique aspect of this retrieval method is its sensitivity to small ice crystals via $\beta_{eff}$.

#### 4 Applying the retrieval method

As mentioned, $\beta_{eff}$ is obtained from the Imaging Infrared Radiometer (IIR) aboard CALIPSO, and is based on effective absorption optical depths retrieved from the 12.05 µm and 10.6 µm channels. IIR retrievals have a resolution of 1 km. A number of selection criteria were applied for the robustness of the retrievals.

Equation 1 was applied only to single-layered semi-transparent cirrus clouds (one cloud layer in an atmospheric column) that do not fully attenuate the CALIOP laser beam, so that the cloud base is detected by CALIOP. The cloud base is in the troposphere and its temperature is required to be smaller than 235 K to ensure that the cloud is entirely composed of ice. Because the relative uncertainties in $\tau_{abs}$ and in $\beta_{eff}$ increase very rapidly as cloud emissivity decreases (Garnier et al., 2013), the lidar layer-integrated attenuated backscatter (IAB) was chosen greater than 0.01 sr$^{-1}$ to avoid very large uncertainties at the smallest optical depths. This resulted in a OD range of about 0.3 to 3.0. Similarly, clouds for which the radiative contrast $R_{BG}$ -$R_{BB}$ between the surface and the cloud is less than 20 K in brightness temperature units are discarded. Finally, IIR observations must be of good quality according to the quality flag.

Our retrievals are related to cloud temperature through the CALIOP attenuated backscatter centroid $T_c$ introduced earlier.

#### 4.1 Relationship between $\beta_{eff}$, $\alpha_{ext}$, IWC, and N

As seen from Eq. (4), (5) and (6), $\beta_{eff}$ and $\alpha_{ext}$ are the two key parameters retrieved from the CALIPSO IIR to derive N/IWC, IWC, and finally N. The interrelationship between $\beta_{eff}$, $\alpha_{ext}$, IWC, and N is illustrated in Fig. 6 (top row), which also shows the range encountered for these properties in the selected cloud population. The red dashed lines are where N = 200 L$^{-1}$, 500 L$^{-1}$ (the liberal and conservative hom thresholds), and 1000 L$^{-1}$. From this we see that both hom and het contribute to cirrus cloud formation. The pink dashed lines are where IWC = 0.5 mg g$^{-3}$, 5 mg g$^{-3}$, or 30 mg g$^{-3}$. Large values of N result from larger values of $\beta_{eff}$ (yielding smaller $D_e$ and much larger N/IWC) and sufficiently large values of $\alpha_{ext}$ so that IWC is sufficiently large for these small values of $D_e$. For our data selection, $\alpha_{ext}$ is mostly between 0.05 km$^{-1}$ and 5 km$^{-1}$. The horizontal red dotted lines for $\beta_{eff}$ < 1.035 indicate where the retrieval is not sensitive to N/IWC. For $\beta_{eff}$ <1.035, N/IWC is set to the maximum possible value (5.6 10$^7$ g$^{-1}$) so that N is a priori overestimated in these conditions. For $\beta_{eff}$ < 1, $D_e$ is set to $D_e$ = 122 µm, as denoted by the horizontal pink lines, and IWC is a priori underestimated for these conditions.



## 4.2 Retrieval uncertainties

The approach to compute the uncertainty in N ($\Delta$N) resulting from the estimated uncertainties in $\beta_{eff}$ and $\alpha_{ext}$ is described in the Appendix. Fig. 6 (bottom row) shows $\Delta$N/N against $\beta_{eff}$ for the same samples as in Fig. 6 (top row). $\Delta$N/N decreases as $\beta_{eff}$ increases, reflecting that the technique is sensitive to small crystals. $\Delta$N/N is found most of

the time < 50% for $\beta_{eff}$ > 1.15. For a given value of $\beta_{eff}$, the variability of $\Delta$N/N is due to the variability of $\Delta\beta_{eff}/\beta_{eff}$ and of $\Delta\alpha_{ext}/\alpha_{ext}$. $\Delta$N/N is found to be larger over land in part because of the sometimes relatively weak radiative contrast. $\Delta\beta_{eff}/\beta_{eff}$ is mostly due to random measurement errors, because systematic errors associated with the retrieval of $\tau_{abs}$(12 µm) and $\tau_{abs}$(10.6 µm) tend to cancel when these are ratioed to calculate $\beta_{eff}$. The uncertainty in $T_{BG}$ contributes more importantly to $\Delta\alpha_{ext}/\alpha_{ext}$ at the smallest emissivities. Uncertainty in $T_{BB}$ is not a major

contributor for semi-transparent clouds of small to medium emissivity.

## 4.3 Comparison with in situ cirrus cloud measurements

Krämer et al. (2009) compiled coincident in situ measurements of N and IWC from 5 field campaigns (10 flights) between 68N and 21S latitude where N was measured by the FSSP probe and IWC was directly measured by various probes as described in Schiller et al. (2008). Krämer et al. (2009) estimated that the FSSP measurements

accounted for at least 80% (but typically > 90%) of the total N in a PSD. These measurements were made at T < 240 K where PSD tend to be relatively narrow and ice particle shattering upstream of particle detection (i.e. the sample volume) is less of a problem (de Reus et al., 2009; Lawson et al., 2008). Moreover, the FSSPs used did not use a flow-straightening shroud in front of the inlet; a practice that will reduce the amount of shattering. The complete data set of in situ IWCs reported in Krämer et al. (2009) extends beyond the 10 field campaigns mentioned

above, and this complete IWC data set is also described in Schiller et al. (2008).

Since this retrieval is sensitive to the smallest ice crystal sizes, it has the advantage of being sensitive to ice nucleation processes, but this also poses certain challenges. For example, the comparison of retrieved and measured N in cirrus clouds is necessarily ambiguous due to (1) the uncertainty in PSD probe measurements at the smallest sizes in a PSD [assuming the probe is capable of measuring N between roughly 5 µm and 50 µm], (2) the PSD size

range used to create the retrieval relationships relative to the PSD size range of the measurements used to test the retrieval, (3) the size range of the retrieved PSD (which is unknown), (4) in situ measurements in optically thin layers below the retrieval limit of the IIR, and (5) the comparison of retrieved layer-averaged N to localized aircraft measurements of N (i.e. the variability in the aircraft measurements at a given temperature is higher than the corresponding variability in the layer averaged retrievals). Regarding (2), since this retrieval was developed from

2D-S probe in situ measurements, ideally it should be validated against 2D-S probe in situ measurements. Comparing with the Krämer et al. (2009) measurements introduces some ambiguity since the smallest size-bin of the 2D-S is from 5-15 µm whereas the Krämer et al. (2009) N measurements are based on the FSSP 100/300 that





sampled particles in the size range 3.0–30/0.6–40 μm diameter, and ice crystals larger than this size range were not recorded. Moreover, the amount of additional uncertainty in the FSSP measurements due to the possibility of shattering was not quantified.

Krämer et al. (2016b) describe a new cirrus cloud data set of in situ measurements of N about a factor of 7 greater in the number of flights relative to the Krämer et al. (2009) N data set described above (10 flights), and these more recent PSD measurements were made with probes designed to minimize the problem of ice particle shattering, using knife-edge inlets or tips to minimize the area susceptible to shattering. Post-processing analysis of ice particle interarrival times (Field et al., 2003) was also used to minimize errors due to shattering; for more details, see Luebke et al. (2016). These results for N are very similar to the N results shown in Krämer et al. (2009), although the new results show somewhat lower values for N for T > 228 K (where PSDs tend to be broader making shattering more likely). The new results for IWC (from 111 flights) were also very similar to the IWC-temperature measurements in Krämer et al. (2009).

Finally, the N/IWC vs. $\beta_{eff}$ and the $1/D_e$ vs. $\beta_{eff}$ relationships shown in Figs. 2 and 3 are assumed to be universally valid. Since they were developed from SPARTICUS and TC4 PSDs (i.e. a limited sampling of mid-latitude and tropical cirrus clouds), these relationships may not be representative for all cirrus clouds sampled world-wide by the IIR. In future work, we will re-examine these relationships using more in situ data from additional field campaigns.

Given the above ambiguities and uncertainties, close agreement between the median retrieved and in situ measured $N(T_c)$ should not be expected, but the temperature-dependence of retrieved and in situ measured N should be similar if the retrieval is valid. The curve fits describing the in situ data of Krämer et al. (2009) are shown in Figs. 7 and 8 by the dashed red curves, and correspond to the maximum, minimum and middle (i.e. mid-point) value of a cloud property as a function of temperature. They are compared with corresponding retrieved mean and median values (solid and dashed black curves, respectively) in these figures. Although the cirrus cloud measurements in Krämer et al. (2009) occurred over both land and ocean, no distinction was made in this regard. But since retrieval uncertainties are greater over land, Figs. 7 and 8 show our retrievals over ocean and land, respectively. Retrieved values are averaged over all seasons for 2013 and over the latitude range roughly corresponding to the field measurements (70N to 25S). Temperature intervals are 4°C. The black dotted horizontal lines in the panel for N correspond to 200 L$^{-1}$ and 500 L$^{-1}$ (i.e. liberal and conservative thresholds for hom).

Since the mass-weighted ice particle size $R_{ice}$ in Krämer et al. (2009) was derived from in situ measurements of IWC/N assuming ice spheres at bulk density (0.92 g cm$^{-3}$), $R_{ice}$ can be inverted to yield in situ measurements of N/IWC. These are compared against our retrieved N/IWC in Figs. 7 and 8. The upper and lower red dashed curves regarding N/IWC in Figs. 7 and 8 were derived from the lower and upper $R_{ice}$ limiting curves in Krämer et al. (2009), respectively.



As shown in the upper left panel, $\beta_{eff}$ can be less than 1.035 or 1.0 (limiting values when calculating N/IWC or $D_e$, respectively) due to instrument noise. In such cases N/IWC or $D_e$ are calculated from these limiting values as described earlier, and this produces the higher sampling densities in the lower regions of Figs. 7 and 8 regarding N/IWC and N. The fraction of samples subject to this procedure is small as indicated in Table 1.

Regarding (4) above, the divergence between the retrieved median and in situ middle value for IWC and N for T < 200 K in Fig. 7 and 8 may be due to the in situ sampled cirrus often having mean layer extinction coefficients smaller than the IIR retrieval limit of about 0.05 km$^{-1}$ (see Fig. 6) resulting from the removal of optical depths below ~ 0.3 from the sampling statistics. Tropical tropopause layer (TTL) cirrus having OD < 0.3 are extensive in the tropics and these cirrus are generally characterized by lower N and IWC (Jensen et al., 2013; Spichtinger and

Krämer, 2013).

In general, the agreement between retrieved and in situ measured quantities in Figs. 7 and 8 is favorable despite the uncertainties involved. Given this agreement, it appears that relative differences in retrieved N, $D_e$ and IWC should be meaningful, and from these relative differences, mechanistic inferences can be made and hypotheses explaining these inferences can be postulated.

**5   Retrieval results and discussion**

**5.1  Frequency of occurrence of selected cirrus samples**

As presented in Sect. 4, the sampled 1-km$^2$ IIR pixels are those for which the atmospheric column contains a single semi-transparent cloud layer of optical depth roughly between 0.3 and 3, of base temperature < 235 K, with a radiative contrast between surface and the cloud of at least 20 K.

Cirrus clouds of optical depth between 0.3 and 3 are geographically widespread across all latitudes and are also in an OD range that makes them radiatively important (Hong and Liu, 2015). Frequency of occurrence is defined as the number of cirrus cloud pixels sampled divided by the number of available IIR pixels. To clarify, a cirrus cloud extending over 20 km horizontally along the lidar track is counted 20 times whereas a cirrus cloud extending over 5 km is counted only 5 times.

Two years of CALIPSO IIR data are considered: 2008 (Dec. 2007 to Nov. 2008) and 2013 (March 2013 to Feb. 2014). It is noted that the version of the GMAO Met data used in the CALIPSO products is not the same in 2008 and in 2013. In 2008, it was GMAO GEOS 5.1 until Sept 2008 and GMAO GEOS 5.2 for Oct 2008 and Nov 2008. In 2013, it is GMAO GEOS FP-IT for the whole period. Retrievals for each month of each year for all latitudes have been analyzed and organized into seasons, with winter as December, January, February (DJF); spring as March,

April, May (MAM); summer as June, July, August (JJA); fall as September, October, November (SON). Figs. 9 and





10 show global maps of the occurrence frequency for all seasons during 2008 and 2013, respectively. The horizontal resolution of these maps is 2° in latitude and 4° in longitude.   Frequency of occurrence is also reported in Table 2 for each season and each 30 degree latitude zone, and for the entire planet during 2008 and 2013. The selection criteria result in very few sampled pixels relative to the number of available IIR pixels, making the frequency of

occurrence generally less than 2%.  Thus what is important in this analysis is not the actual frequency value but the relative differences in these values with respect to season and latitude. It is seen that despite our cloud subsampling, the geographical distribution of the occurrence frequencies is consistent with previous findings for ice clouds (T < 0°C) of optical depth between 0.3 and 3 (Hong and Liu, 2015). The greatest occurrence frequencies are in the tropics (i.e. 30° S-30° N) and are associated with anvil cirrus from deep convection and relatively optically thick TTL

cirrus. The occurrence frequency during Arctic (i. e. 60° N-82° N latitude zone ) winter is more than twice the frequency of other Arctic seasons.  In the Antarctic (i.e. the 60° S-82° S latitude zone), frequency of occurrence is greatest in the spring and second-greatest during winter, in agreement with previous studies (Nazaryan et al., 2008; Hong and Liu, 2015). This is important since at high latitudes, the net radiative effect of ice clouds is strongest during the "cold season" where solar zenith angles are relatively low and ice cloud amount is relatively high (Hong

and Liu, 2015).  Therefore, the cirrus cloud formation mechanism that governs cirrus microphysical properties will be important at high latitudes during winter and also during spring for the Antarctic region.

## 5.2   Seasonal maps of hom likelihood

Figures 11 and 12 for years 2008 and 2013, respectively, show global maps of the fraction of the selected cirrus pixels having N > 500 L$^{-1}$; indicated by the color bar, for each season. Each 2°×4° grid that is indicated as cloudy

contains at least 15 1-km$^2$ cirrus cloud samples to yield meaningful statistics.  These fractions are averaged zonally in Fig. 13 (left column) by separating land and ocean, for all four seasons in 2013. The vertical bars represent our uncertainty estimate, derived from the fractions of samples with N-$\Delta$N >500 L$^{-1}$ and N+ $\Delta$N>500 L$^{-1}$. As discussed in Sect. 4.2, the relative uncertainty, $\Delta N/N$, is generally < 50% when $\beta_{eff}$ > 1.15.  This is illustrated in Fig. 13 (right column), which shows the fraction of samples with N $\pm$ $\Delta$N > 500 L$^{-1}$ and $\beta_{eff}$ > 1.15.  It is seen that the zonal

fractions in the right and left column of Fig. 13 are similar despite the larger uncertainties when all values of $\beta_{eff}$ are included, lending greater confidence to the Fig. 11 and 12 results.

Over land, there is a marked increase in the fraction of samples with N>500 L$^{-1}$ poleward of 30°N and 30°S latitude, especially during the winter season, and it appears associated with mountainous terrain (see Fig. 14, which is an elevation map of the Earth).  This is supporting our classification of cirrus having N> 500 L$^{-1}$ as hom cirrus. Indeed,

this is consistent with our knowledge regarding hom since hom is most sensitive to the cooling rate (i.e. the updraft) and also depends on the concentration of IN.  Mineral dust concentrations (a principal IN) at these latitudes at 200 hPa are predicted to be low during winter relative to other seasons (Storelvmo and Herger, 2014), and mountain




induced wave clouds can provide relatively strong updrafts. Perhaps less expected is the broad coverage of hom cirrus associated with mountainous terrain. While lenticular wave clouds are quite limited in areal extent and hence relatively insignificant to cloud radiative forcing globally (e.g. Krämer et al., 2016a), the hom cirrus associated with mountains in Figs. 11 and 12 exhibit broad areal coverage and thus may have an impact on climate, as will become

more apparent in Sect. 5.3. The Upper Atmosphere Research Satellite Microwave Limb Sounder (UARS MLS) has been used to detect stratospheric gravity waves over Antarctica (Wu and Jiang, 2002) and mountain waves over the Andes of South America (Jiang et al., 2002) and over specific mountainous terrain in the Northern Hemisphere (see Jiang et al., 2004). The Northern Hemisphere MLS variance enhancements associated with these mountain-induced waves peak during the winter season, while in the Southern Hemisphere they peak during the spring (SON). The

locations and seasonality of these wave perturbations coincide well with the locations/seasons associated with widespread hom cirrus in Figs. 11 and 12. Brightness temperature perturbations and variations detected using the AIRS/Aqua 4.3 µm channel have also been used to study gravity wave activity at gravity wave hotspots in the Southern Hemisphere (Hoffmann et al., 2016). These results strongly support the above noted studies regarding Antarctica and the Andes.

One might argue that the relatively high N in eastern Asia results from mineral dust transported from known dust sources; the Kara-Kum Desert just east of the Caspian Sea, the Taklimakan Desert in extreme western China and the Mongolian desert. However, N in this region is highest during winter and lowest during summer, which is out-of-phase with predicted dust concentrations at 200 hPa (Storelvmo and Herger, 2014). Moreover, these regions receive precipitation mainly in winter and early spring, often in the form of snow, thus limiting the production of mineral

dust during winter.

Regarding the location of het cirrus, small fractions of samples with $N > 500$ L$^{-1}$ indicate a predominance of het cirrus. Het cirrus appear most abundant over the oceans, in the tropics and over low-elevation land regions poleward of 30° N and 30° S latitude. Outside the tropics, these are relatively flat regions where atmospheric wave amplitudes are likely to be low relative to mountain-induced wave amplitudes. Within the tropics-dominated ±30°

zone, het cirrus associated with deep convection appear to dominate over both oceans and mountains as observed over the Andes Mountains of South America in Figs. 11 and 12.

As discussed in Krämer et al. (2016a) and Luebke et al. (2016) and in Sect. 1, anvil cirrus can be described as liquid origin cirrus. Liquid origin cirrus are associated with larger ice crystals and higher IWC relative to in situ cirrus. Moreover, liquid origin or anvil cirrus clouds form in the presence of pre-existing ice particles. With pre-existing

ice, het is strongly favored since a substantial amount of ice surface area is already present, usually precluding the RHi needed for hom to occur (Shi et al., 2015; Zhou et al., 2015). This may be a primary reason for the relatively low N in the tropics where anvil cirrus clouds prevail. Another factor is that boundary layer air, potentially rich in





IN, is advected to cirrus levels during deep convection. The combined effect of pre-existing ice and IN enrichment from deep convection appears to prevent the RHi from reaching the hom threshold, even under the high updraft conditions expected over the Andes Mountains in the tropics that abruptly rise as high as 6000 m or more. To summarize, anvil cirrus are considered liquid origin cirrus having pre-existing ice that produces conditions favoring

heterogeneous nucleation processes.

The impact of IN on N may be evident over the southern oceans (30° S-60° S) where IN concentrations are predicted to be low (Storelvmo and Herger, 2014). Figures 11 and 12 show higher N fractions over the southern oceans relative to the tropics (including tropical oceans), and this may indicate that hom is more active over the southern oceans (due to lower IN at cirrus levels) relative to the tropics.

**5.3 Quantitative analysis as a function of altitude and latitude**

To evaluate the fraction of $N > 500 \text{ L}^{-1}$ as a function of altitude we look at the distribution of CALIOP centroid altitudes for our selected cloud population. This section is dedicated to characterizing the vertical distribution of IIR retrieval clouds for the tropics, mid-latitudes and the high latitudes, and for all four seasons at each latitude zone, by averaging the years 2008 and 2013. This information is given in Figs. 15-18, where each figure represents one

season showing all six latitude zones (60° N-82° N, 30° N-60° N, 0° N-30° N, 30° S-0° S, 60° S-30° S, and 82° S-60° S). The left column of each figure is for ocean only, the middle column is for land only, and the right column is for ocean and land area combined. Within each panel there are four histograms of IIR 1-km² pixel counts as a function of the CALIOP 532 nm layer centroid altitude, with ½ km vertical increments. The black histogram shows the number of sampled cirrus cloud. The number of sampled pixels having $N > 500 \text{ L}^{-1}$ is shown by the orange

histogram. The red histogram gives the number of samples having $N > 500 \text{ L}^{-1}$ and $\beta_{eff} > 1.15$, which indicates uncertainty $\Delta N/N$ is generally < 50% and that $D_e < 45$ μm. Finally, the blue histogram gives the number of samples having $\beta_{eff} > 1.15$ (i.e. $D_e < 45$ μm). The area under each colored histogram divided by the area under the black histogram yields the corresponding overall fraction. Thus, the fraction of cirrus having $N > 500 \text{ L}^{-1}$, $N > 500 \text{ L}^{-1}$ and $\beta_{eff} > 1.15$, and $\beta_{eff} > 1.15$ is indicated by the orange, red, and blue numbers under "Fraction", respectively.

Following our convention that $N > 500 \text{ L}^{-1}$ corresponds to hom cirrus, the hom cirrus histogram (orange) closely tracks the red histrogram, indicating relatively low N uncertainties and small ice particle sizes are associated with hom cirrus. Outside the tropics (±30°), the blue histogram tracks the orange histogram fairly closely, indicating that cirrus PSDs having $D_e < 45$ μm tend to be associated with hom cirrus.

In the Northern Hemisphere high latitudes (60° N-82° N), sampled cirrus occur mostly over land, with a peak during

winter. The fraction of hom cirrus is generally greater over land than ocean, although this reverses during summer, and the fraction over ocean is greater than at lower latitudes. Averaging over all seasons and both years, the hom





cirrus fraction for land and ocean combined is 0.433±0.046 (± standard deviation), a fractional variability of only 11%. This relatively high hom fraction and low variance may be due to lower IN concentrations in the Arctic throughout the year.

In the Northern Hemisphere mid-latitudes (30° N-60° N), sampled cirrus also occur more frequently over land, but since more of the Earth is covered by ocean at these latitudes, cirrus are sampled more frequently over ocean here relative to oceans at 60° N-84° N. Cirrus cloud coverage peaks during the spring. The fraction of hom cirrus is considerably greater over land than ocean during all seasons, peaking during the winter (DJF) with values of 0.43 for land only, 0.25 for ocean only, and 0.37 for land and ocean combined. At these latitudes mountains appear responsible for the greater hom cirrus fraction over land, as discussed earlier. Gravity waves could also be

responsible for most of the hom cirrus over ocean, albeit generated from adjustments in unbalanced flows in jet-streams and frontal systems (Fritts and Alexander, 2003; Wu and Zhang, 2004; Plougonven and Zhang, 2014) or convection (e.g. Alexander and Pfister, 1995; Vincent and Alexander, 2000).

In the tropics and subtropics between 30° N and 30° S, sampled cirrus occur mostly over the ocean with less seasonal dependence in their coverage relative to the Arctic. The fraction of hom cirrus is slightly greater over land,

and for land and ocean combined, the mean fraction for both years and all seasons is 0.117±0.015. In addition to the hom fraction, this region also differs from the mid- and high latitudes in that the number of hom cirrus samples tends to peak at lower altitudes (relative to the altitude range for cirrus occurrence) while the blue histogram for cirrus having $D_e < 45$ μm does not track the hom cirrus histogram at higher altitudes but rather contributes more and more to the total sampled cirrus with increasing altitude. At the highest altitudes in the TTL region, most of the cirrus

samples have $D_e < 45$ μm. Note that most of the TTL cirrus are not detected by this retrieval method ($0.3 \le OD \le 3.0$).

These results may be detailed enough to test some of our ideas on cirrus cloud formation. For example, based on 15 flights measuring cirrus clouds over the continental USA, Diao et al. (2015) found that the ice nucleation zone was generally near cloud top near the thermal tropopause. They point out that this may occur for two reasons: (1) abrupt

changes of temperature and humidity near the tropopause, and (2) frequent occurrence of clear-air turbulence around the jet-stream. Regarding (1), the Clausius-Clapeyron equation predicts that a small perturbation of the moisture or temperature field near the thermal tropopause will generally produce a higher RHi than would occur at lower levels. Thus, as temperatures decrease, the RHi threshold for hom is more likely to be reached. In the northern mid-latitudes during winter over land, there appears to be evidence of this, where the orange histogram indicates that

hom cirrus generally contribute to the total in greater percentages at the highest altitudes.

During summer over land at these mid-latitudes, just the opposite is evident, with the number of hom cirrus samples and the number of small $D_e$ cirrus samples peaking at lower altitudes (relative to the altitude range for cirrus



occurrence). Due to the close correspondence of the orange and blue histograms, it appears that hom or another process that produces high N (like ice multiplication processes associated with deep convection; see Lawson et al., 2015) is primarily responsible for $D_e < 45$ µm. At higher altitudes (above these peaks), ice particle sizes and concentrations appear to increase and decrease, respectively.

Over oceans in the tropics (±30°), the histogram behavior differs from mid-latitude summer over land (also characterized by deep convection), with samples having $D_e < 45$ µm contributing progressively higher percentages to the total with increasing altitude. Conversely, samples having $N > 500$ L$^{-1}$ peak at lower altitudes. The following discussion offers a possible explanation for the anti-correlation between the orange and blue histogram curves that begins ~ 13.5 km. This behavior might have nothing to do with hom, but rather ice multiplication processes as

described in Lawson et al. (2015). In that study it was found that for deep convection over ocean with updrafts typically on the order of 7 to 10 m s$^{-1}$, rapid glaciation occurs from well documented ice multiplication processes between -12° and -20°C that are capable of producing $N > 500$ L$^{-1}$. These ice particles can be easily advected from the "glaciation zone" into the "cirrus zone" and detrained at a strong inversion layer to form anvil cirrus. Not all deep convection in the tropics reaches the tropopause, with anvils forming well below the tropopause. Widespread

in situ cirrus typically form above the anvils in the TTL region and these cirrus are characterized by relatively low N and low IWCs (Spichtinger and Krämer, 2013; Krämer et al, 2016a). Moreover, observed frequency distributions of N in these TTL cirrus can be explained through the superposition of high-frequency internal gravity waves with very slow large-scale motions, with hom accounting for ~ 79% of N and a combination of het and hom accounting for ~ 20% of N. The relatively low N predicted is due to the shortness of the gravity waves, which stalls freezing events

before a higher ice crystal concentration can be formed. This, along with very low water vapor concentrations, may also limit the size of the ice crystals, possibly explaining the higher abundance of samples having $D_e < 45$ µm at the higher altitudes. This transition from anvil to TTL cirrus can be seen in CALIOP extinction measurements (Gasparini et al., 2016, Fig. 7e), where a strong extinction maximum exists ~ 203 K (13.5 km) indicating the top-level for anvil cirrus. Above this level, cirrus extinction is much weaker, extending up to the tropopause near 193 K.

Evidently the cirrus between 203 K and 193 K correspond to in situ cirrus, and this may account for the anti-correlation we observe ~ 13.5 km regarding the orange and blue histogram curves.

Over land in the tropics (±30°), the noted anti-correlation between the orange and blue histograms is less pronounced, and the fraction of samples having $N > 500$ L$^{-1}$ is slightly greater than over oceans.

In the Southern Hemisphere mid-latitudes (30° S-60° S) there is mostly ocean and thus most cirrus are found over

the ocean. However, over land during winter (JJA), the histogram patterns resemble those of the Northern Hemisphere mid-latitudes, with hom cirrus prevailing at the highest altitudes. Again, this supports the cirrus formation mechanisms suggested by Diao et al. (2015). During summer (DJF) over land, the pattern is again similar





to that found in the Northern Hemisphere mid-latitudes during summer. On average, the hom fraction for the southern oceans is 0.267±0.026, whereas in the Northern Hemisphere over ocean, it is 0.216±0.024. This difference might result from lower IN concentrations in the Southern Hemisphere.

In the Southern Hemisphere high latitudes (60° S-82° S) the amount of cirrus clouds sampled over the ocean is
almost negligible, and cirrus are most abundant over land during winter (JJA) and spring (SON). On average, the fraction of hom cirrus for land and ocean combined is 0.501±0.062, which seems consistent with the lower IN concentrations and mountainous terrain. During spring and sometimes other seasons, the histogram patterns resemble those of the Northern Hemisphere mid-latitudes, with hom cirrus contributing the most to cloud coverage at the highest altitudes. During winter and spring some cirrus clouds over Antarctica extended into the stratosphere
up to 25 km altitude (although their centroid altitude was much lower). Their elevated centroid altitudes are shown by the extended tail of the histograms in Figs. 17 and 18.

### 5.4  Effective diameter

The radiative impact of the above noted changes in cloud properties with season, latitude and surface condition will depend in part on their temperature dependence. Cloud radiative properties in climate models are generally
determined by the cloud IWC and $D_e$. As discussed in Sect. 4.1, hom nucleation (N> 500 L$^{-1}$) will manifest primarily through $\beta_{eff}$, and therefore through $D_e$, providing that IWC is sufficiently large. However, no coherent relationship was consistently found between IWC and the apparent nucleation mechanism for the cirrus sampled here.

Figures 19 and 20 show global maps of the median $D_e$ for the CALIOP centroid $T_c$ interval 206-218 K during 2008
and 2013, respectively. Over land during winter, poleward of 30° N and 30° S latitude, $D_e$ is often less than ~ 45 μm. This is consistent with hom being common there during winter since hom tends to produce many small ice crystals relative to het. This also occurs during spring and fall to a lesser extent. During summer $D_e$ is not much different in this region than it is in the tropical regions, except for the tip of South America, Antarctica and perhaps poleward of 60° N latitude (where relatively few cirrus samples are available in this temperature zone). This may be
due to deep convection moving to higher latitudes during summer.

In the tropics over land and ocean, $D_e$ tends to be largest. As mentioned, this may be due to anvil cirrus being a type of liquid origin cirrus associated with pre-existing ice that suppresses RHi.

Over the southern oceans, $D_e$ appears slightly smaller than $D_e$ over the tropics, especially during the winter season. This is consistent with the slightly higher N fraction over the southern oceans, as noted above.





The $D_e(T_c)$ profiles are shown for the winter and summer of 2013 in the Northern Hemisphere for different latitude zones over ocean and land in Figs. 21 and 22. The mean and median $D_e$ are given by the solid and dashed curves, respectively, and the vertical lines are standard deviations. The color bar gives the sampling density normalized by the maximum number of samples in log scale. The band of high sampling density near $D_e = 120$ μm is due to

retrievals of $\beta_{eff} < 1.0$ (partially due to measurement uncertainties and IIR instrument noise).

Comparing $D_e$ profiles north of 30° N latitude in winter over land with $D_e$ profiles during summer for these same latitudes, it is evident that significant seasonal changes in $D_e$ occur over the entire cirrus temperature domain. These $D_e$ changes due to changes in the contribution from hom cirrus can be on the order of 15 to 20 μm and thus may have a significant impact on cirrus radiative properties and climate (taking into account the seasonal changes in

cirrus cloud coverage in this region).

### 5.5   Working hypotheses for global cirrus cloud formation

Sect. 5.2, 5.3, and 5.4 show a pronounced seasonal cycle in the Northern Hemisphere mid-latitudes over land in terms of the fraction of hom cirrus, with higher N and smaller $D_e$ during the winter season. We postulate that this is partially due to the seasonal cycle of deep convection, with deep convection (1) replenishing the supply of IN at

cirrus levels, and (2) producing anvil cirrus that form in the presence of pre-existing ice (which suppress supersaturations). This should favor het cirrus when cirrus are formed in situ, and the pre-existing ice associated with anvil cirrus should result in anvil cirrus characterized by relatively low N and high $D_e$. Hom cirrus are common over mountainous terrain during the boreal winter north of 30° N since deep convection is relatively absent, the troposphere is more stratified with lower IN concentrations at cirrus levels, and mountain-induced waves

yield strong and sustained updrafts at cirrus cloud levels (allowing RHi to reach the hom threshold). Such waves during summer may be diminished due to a weaker jet-stream and calmer cirrus-level winds. In addition, much of the land at high latitudes during winter is covered by snow, resulting in lower mineral dust IN.

Over oceans outside the tropics, het appears to prevail. Due to the relatively smooth ocean surface, lower amplitude atmospheric waves (relative to mountain-induce waves) at cirrus cloud levels are expected. This may limit the RHi

within these waves, making it more difficult to attain values needed to initiate hom.

Within the latitude zone of 30° N to 30° S, the ice nucleation mechanism (Figs. 11 and 12) does not appear to be sensitive to surface conditions (i.e. land vs. ocean). This could be largely due to the dominance of anvil cirrus from deep convection in this region, where pre-existing ice generally keeps RHi below the hom threshold during anvil formation. Deep convection will also replenish the IN at cirrus levels so that in situ cirrus tend to be het cirrus.





There is relatively little land between 30° S and 60° S, with most of it associated with South America and Australia (a rich source of mineral dust; Gasparini and Lohmann, 2016). Land in this latitude zone appears to exhibit a similar hom-het annual cycle to North America but with a stronger hom signature in South America probably due to (1) mountain-induced waves from the Andes Mountains and (2) lower IN concentrations in the Southern Hemisphere.

Over pristine Antarctica, the terrain is high and often mountainous near the coast, and IN concentrations are expected to be minimal (Storelvmo and Herger, 2014), with both factors favoring hom cirrus. Accordingly, over Antarctica cirrus clouds exhibit relatively high N and small $D_e$ throughout the year.

## 6   Connections with other studies

### 6.1   Ice nucleation studies

As several GCM modeling studies that use the pre-existing ice assumption have convincingly shown (Shi et al., 2015; Penner et al., 2015; Zhou et al., 2015; Gasparini and Lohmann, 2016), pre-existing ice often prevents the RHi in a cirrus cloud updraft from reaching the hom threshold, thus resulting in het cirrus or "liquid origin cirrus" such as anvil cirrus (Krämer et al., 2016a; Luebke et al., 2016). In these GCM studies, this assumption was applied to all cirrus clouds worldwide. In our remote sensing study, the fraction of N > 500 $L^{-1}$ is lowest in the tropics (Fig. 11

and 12) and primarily corresponds to anvil cirrus with pre-existing ice (see Sect. 1). In regions where hom prevails (fraction exceeds 0.5), N is much higher than in these tropical regions, which is unlikely to occur in the presence of pre-existing ice over a moderate range of updraft speeds (Shi et al., 2015; Zhou et al., 2016). In addition to this study, the observational findings of Diao et al. (2015) described in Sect. 5.3 suggest that the pre-existing ice assumption may not be appropriate for in situ cirrus since the nucleation zone is near cloud top where little ice

surface area exists to reduce RHi.

Other observational studies (Diao et al., 2013; Diao et al., 2014) have shown that in situ cirrus clouds evolve in stages that can be described as nucleation, early ice crystal growth, later growth and sedimentation/sublimation, with the nucleation stage preceded by a clear-sky region of supersaturation with respect to ice (ice supersaturation region or ISSR). While the ISSR and sedimentation stages are long-lived, the ice nucleation stage is relatively short-lived.

This implies that in situ cirrus initially form without the presence of pre-existing ice, but rather result from a clear-sky ISSR. Thereafter the cirrus can exist for long periods during their sedimentation stage. To summarize, with regards to in situ cirrus clouds at mid-latitudes, our results appear consistent with the findings of Diao et al. (2013, 2014, 2015) and do not imply the existence of pre-existing ice.

The findings of this study are also consistent with the findings of Cziczo et al. (2013) which, based on the four field

campaigns studied, showed that het was the freezing mechanism in 94% of their cirrus cloud encounters. This





agreement can be understood if one considers the locations and seasons during which the field campaigns studied by Cziczo et al. took place. Two campaigns were near Costa Rica; one during January-February and one during July-August, while another was in southern Florida during July and another was in the south-central USA during March-April. Figures 11 and 12 show that het cirrus conditions appear common in the regions/seasons during which these

field campaigns were conducted.

A recent field study by Voigt et al. (2016) has sampled mid-latitude cirrus over Europe using a sampling method similar to that used by Cziczo et al. (2013). Both natural and contrail cirrus were present during sampling, with results indicating a predominance of hom cirrus. The sampling method provides detailed mechanistic evidence corroborating our inferences of hom and het cirrus in this region.

## 6.2  GCM studies

At least two GCMs predict the supersaturation of water vapor with respect to ice; the Community Atmosphere Model version 5 (CAM5; see Gettelman et al., 2010) and the ECHAM-HAM GCM (Zhang et al., 2012; Kuebbeler et al., 2014). These models realistically treat competition effects between het and hom (e.g. Liu and Penner, 2005; Kärcher et al., 2006; Barahona and Nenes, 2009) and have the ability to predict the geographic locations of hom-

and het-dominated cirrus clouds. Many factors determine the relative roles of hom and het in cirrus formation (e.g. Liu et al., 2012; Zhang et al., 2013; Zhou et al, 2015), and various configurations of these "ice nucleation factors" in CAM5 and ECHAM5/6 will yield various predictions for het and hom contributions. One example is illustrated in Fig. 3 of Penner et al. (2015) where pre-existing ice is assumed, and where IN contributions include mineral dust and 0.1% of the secondary organic aerosol. In this case hom dominates in the tropics (especially the tropical

Pacific) and in most of the Southern Hemisphere, whereas het dominates outside the tropics in the Northern Hemisphere. These results differ from this CALIPSO study in both the tropics and mid-latitudes as shown in Figs. 11 and 12, and in Sect. 5.3.

Gasparini and Lohmann (2016) used the ECHAM6 GCM to produce a global map at 200 hPa showing annual averages of the percent contribution of het, hom and detrainment (from deep convection) to ice crystal production

(their Fig. 2) and to produce zonal means as a function of latitude and temperature of these percent contributions (their Fig. 3). Pre-existing ice was assumed worldwide and mineral dust was the primary IN. Their results agree well with our results in the tropics where virtually all ice was produced through detrainment or het. Outside the tropics at 200 hPa there is also qualitative agreement with our results in that hom cirrus are associated with mountainous terrain in approximately the same places. However, below 200 hPa, this agreement is lost as detrained

ice and het dominate (their Fig. 3). A more detailed comparison between this GCM study and our results is difficult since seasonal variations were not reported in this ECHAM6 study. For example, during winter, Figs. 11 and 12 show that hom cirrus outside the tropics extend over a much broader geographical area than shown by the ECHAM6





results at 200 hPa. This is especially evident over Greenland and Antarctica where het cirrus in ECHAM6 dominate everywhere except along the coastline where there is an abrupt change in altitude. Moreover, Fig. 3 in Gasparini and Lohmann indicates that detrained ice and het strongly dominate ice formation in cirrus clouds (T < -35°C) below the 200 hPa level at middle- and high-latitudes, whereas this study (Sect. 5.3) shows that hom contributes

significantly even at relatively low cirrus levels. Overall, the Gasparini and Lohmann study concludes that het and detrained ice strongly dominate the cirrus cloud net radiative forcing, even at high latitudes, whereas the results of this CALIPSO study indicate that hom should have a greater radiative impact at high latitudes (recall that the N > 500 L$^{-1}$ threshold for hom cirrus is a conservative threshold, and that a liberal threshold for hom cirrus is N > 200 L$^{-1}$. Thus the fraction of hom cirrus in the polar regions could exceed 50%).

While some limitations of these studies have been addressed (Penner et al., 2015; Gasparini and Lohmann, 2016), the CAM5 GCM studies by Storelvmo and Herger (2014) and Storelvmo et al. (2014) used predicted concentrations of mineral dust to estimate the seasonal and latitude dependence of hom cirrus, resulting in the prediction that hom cirrus prevail at high latitudes. Their findings are supported by this remote sensing study.

## 7   Summary and conclusions

This research was born out of recognition that a practical understanding of ice nucleation in cirrus clouds was being hampered by an inability to globally observe the cirrus cloud ice particle number concentration (N) as a function of temperature (or altitude), latitude, season and surface type. A new satellite remote sensing method, which is sensitive to N over the range that typically characterizes a transition from het to hom, was developed to address this need. This was made possible by exploiting the fact that most of the cloud emissivity difference between the split-

window channels at 11 and 12 μm is due to wave resonance absorption; a process sensitive to the smallest ice crystals that dominate N (Mitchell et al., 2010). Due to this process, a tight relationship between N/IWC and $\beta_{eff}$ was obtained over the region where a transition between het and hom generally occurs (see Fig. 2). This relationship, and a similar tight relationship between $D_e$ and $\beta_{eff}$, are the unique aspects of this retrieval and make it self-consistent through the shared dependence on $\beta_{eff}$. Although the retrieval is restricted to single-layer cirrus cloud

optical depths between about 0.3 and 3.0, this optical depth range is likely to be the most radiatively significant range due to the lower cirrus cloud frequency of occurrence at higher OD and a much lower cirrus cloud mean emissivity at the lower ODs (Hong and Liu, 2015). In other words, for the sampled single layer clouds, the cirrus clouds that the IIR senses best in the window channels will also have the most influence on the Earth's longwave radiation budget.

A two-year global and seasonal analysis of these CALIPSO observations that uses N to discriminate between hom and het cirrus indicates that hom cirrus are common during winter north of 30° N latitude over mountainous terrain. The same is true in the Southern Hemisphere although there is much less land mass south of 30° S. Over the oceans



at all latitudes, het cirrus are dominant to varying degrees, and in the tropics (±30°lat.), het cirrus prevail over land. On average, hom cirrus are found 43% of the time in the Arctic and 50% of the time in the Antarctic. Hypotheses were proposed to explain these results.

Future work will continue retrieval validation efforts, and will investigate the radiative implications of these retrievals, including the potential impact of Arctic winter cirrus on the meridional temperature gradient as discussed below in Sect. 7.1.

### 7.1   A potential link between Arctic cirrus and mid-latitude weather

These retrieval results indicate that at high latitudes there tends to be the greatest cirrus cloud coverage during winter in the Arctic and during spring (SON) in the Antarctic (where relatively high N and small $D_e$ occur throughout the year in both regions). While this study only considers a subset of cirrus clouds and two years of retrievals, our findings on the seasonal dependence of cirrus cloud coverage are consistent with other satellite cirrus cloud studies that consider a broader range of conditions over longer periods (e.g. Nazaryan et al., 2008; Hong and Liu, 2015). Independent of the macro- and microphysical cirrus cloud attributes found in this study, at high latitudes there are important seasonal changes to the cirrus cloud shortwave (SW) and longwave (LW) radiative forcing due to a changing solar zenith angle, with the SW and LW components almost cancelling during summer but during winter, the LW component strongly prevails, producing a strong net warming at the top of atmosphere (TOA) and at the surface (Hong and Liu, 2015; Storelvmo et al., 2014). This indicates that the strongest net radiative forcing by cirrus clouds on Arctic (Antarctic) climate occurs during winter (spring). This seasonal cycle of the solar zenith angle combined with the unique macro- and microphysical properties of Arctic cirrus during winter suggests that wintertime Arctic cirrus may have a significant warming effect on Arctic climate. A satellite remote sensing study of ice clouds (T < 0°C) by Hong and Liu (2015) found that at high latitudes, ice cloud net radiative forcing at the TOA and at the surface during the cold season is > 2 W m$^{-2}$ for a cirrus cloud OD of 1.5. Since the most severe effects of global warming occur at high latitudes, it is critical to understand the factors controlling the occurrence of het and hom cirrus in this region as well as cirrus cloud coverage in this region.

A potential link to mid-latitude winter weather is the possible impact of the winter Arctic cirrus on the meridional (north-south) temperature gradient between the Arctic and mid-latitudes. The cirrus-induced winter warming described above will occur throughout the troposphere (Chen et al., 2000; Hong and Liu, 2015), and will thus act to reduce this temperature gradient in the upper troposphere (UT). While it is not clear how this would impact weather, some type of impact is likely if the warming is significant, and several possible scenarios are described in Cohen et al. (2014) and Barnes and Screen (2015). While many papers have been published recently regarding potential effects of Arctic Amplification (henceforth AA; the observation that the mean Arctic temperature rise due to greenhouse gases is at least a factor of two greater relative to the adjacent mid-latitudes) on mid-latitude weather,





it is important to note that AA due to the loss of sea ice and snow cover primarily affects low-level temperatures while AA due to winter cirrus strongly affects the UT. A theoretical link between AA and the jet-stream is found in the thermal wind balance, which states that a reduced meridional temperature gradient tends to produce a reduced vertical gradient in the zonal-wind field, depending on other factors like changes in surface winds, storm tracks and

the tropopause height (Barnes and Screen, 2015). Thus AA could lead to a weaker jet-stream having more amplified Rossby waves and associated extreme weather events as hypothesized by Francis and Vavrus (2012; 2015), but it is currently not clear whether such a phenomenon is occurring or will be occurring (Barnes and Screen, 2015).

As described in Barnes and Screen (2015), GCM simulations from the fifth Coupled Model Intercomparison Project (CMIP5) show that while the lower troposphere during Arctic winter is projected to warm substantially by 2100, this

is not happening in the Arctic UT where little warming is projected. Moreover, in the tropics the models predict the strongest warming in 2100 occurs in the UT. These effects decrease the meridional temperature gradient at low levels and increase the temperature gradient in the UT. These low- and high-level gradients have competing effects on the jet-stream, with a decreasing low-level gradient acting to weaken the jet-stream and shift it towards the equator, while an increasing UT gradient acts to strengthen the jet-stream and shift it poleward (Barnes and Screen,

2015). An interesting question to ask here is whether the CMIP5 GCMs adequately describe the changes in winter Arctic cirrus that satellite remote sensing studies observe. If they do not, and the winter heating from Arctic cirrus clouds is underestimated in the models, then the meridional UT temperature gradient may be overestimated during winter. If this were the case, then increasing Arctic cirrus coverage during winter in the models would tend to weaken the simulated jet-stream and shift it further towards the equator. Future GCM research should determine

whether predicted cirrus cloud coverage and microphysics is consistent with the results from satellite studies such as this one, and strive for consistency with these remote observations. Then it could be determined whether the UT heating from the winter Arctic cirrus would be enough to produce significant changes in the simulated Northern Hemisphere mid-latitude circulation.

A related question is whether wintertime Arctic cirrus are increasing, causing a change in jet-stream behavior.

Screen et al. (2012; 2015), as well as other studies (Ding et al., 2014; Perlwitz et al., 2015), give evidence that AA is due to both local and remote effects. Remote effects outside the Arctic include changes in tropical (Ding et al., 2014) and mid-latitude (Screen and Francis, 2016) SSTs and Rossby waves that enhance the transport of energy and moisture northwards into the Arctic, such as storm systems along the storm track. Both Screen et al. (2012) and Francis and Vavrus (2015) found evidence of increased remote energy transport into the Arctic, especially after

2000. This occurred mostly during the fall (Francis and Vavrus, 2015). This transport may have contributed to the observed buildup of Arctic cirrus clouds during winter when temperatures plummet. In this way remote effects may enhance Arctic winter cirrus and their associated heating rates, which may affect jet-stream dynamics. Future work will report on seasonal trends in Arctic cirrus frequency of occurrence.



**Appendix: Retrieval uncertainty analysis**

We begin this analysis with our retrieval equation for the ice particle number concentration:

$$N = \frac{\rho_i}{3} \times \frac{\left[2/\overline{Q}_{abs}(12\mu m)\right]\tau_{abs}(12\mu m)}{\Delta z_{eq}} \times D_e \times \left(\frac{N}{IWC}\right) \tag{A1}$$

with $\rho_i = 0.917 \times 10^6$ g m$^{-3}$. The quantities $D_e$ and N/IWC are retrieved from $\beta_{eff}$ using the regression curves in Figs. 2 and 3, respectively. By writing x= $\beta_{eff}$, N/IWC is expressed as

$$\left(\frac{N}{IWC}\right)(g^{-1}) = 10^9(a2.x^2 + a1x + a0) \tag{A2}$$

with a2=2.10828, a1=-3.93097, and a0=1.81064, and

$$D_e(\mu m) = (b2.x^2 + b1x + b0)^{-1} \tag{A3}$$

with b2=0.00751586, b1=0.0777754, and b0=-0.0770823. We now define:

$$f(x) = 10^{-6}\left(\frac{N}{IWC}\right)(g^{-1}) \times D_e(\mu m) \times \frac{1}{3.27} = \frac{a2.x^2 + a1x + a0}{b2x^2 + b1x + b0}.\times \frac{10^3}{3.27} \tag{A4}$$

so that Eq. (A1) can be re-written as:

$$N(L^{-1}) = f(x) \times \alpha_{ext}(km^{-1}) \tag{A5}$$

with

$$\alpha_{ext} = \frac{2/\overline{Q}_{abs}(12\mu m).}{\Delta z_{eq}} \times \tau_{abs}(12\mu m) \tag{A6}$$

Assuming a negligible error in $\Delta Z_{eq}$, and writing $\tau_{abs}$(12 μm) as $\tau_{12}$ and $\tau_{abs}$(10.6 μm) as $\tau_{10}$ for more clarity, so that x=$\tau_{12}/\tau_{10}$, the derivative of N can be written:

$$\frac{dN}{N} = \frac{1}{f}\frac{\partial f}{\partial x} \cdot x \cdot \left(\frac{d\tau_{12}}{\tau_{12}} - \frac{d\tau_{10}}{\tau_{10}}\right) + \frac{d\tau_{12}}{\tau_{12}} \tag{A7}$$



Errors in $\tau_{12}$ and in $\tau_{10}$ are computed by propagating errors in i)the measured brightness temperatures $T_m$, ii) the background brightness temperatures $T_{BG}$, and iii)the blackbody brightness temperatures $T_{BB}$ (Garnier et al., 2015).

The uncertainties in $T_{m10}$ at 10.6 μm and in $T_{m12}$ at 12.05 μm are random errors set to 0.3 K according to the IIR performance assessment established by the Centre National d'Etudes Spatiales (CNES) assuming no systematic bias in the calibration. They are statistically independent.

Because the same cloud temperature is used to compute $\tau_{12}$ and $\tau_{10}$, the uncertainty $\Delta T_{BB}$ is the same at 10.6 and at 12.05 μm. A random error of +/-2K is estimated to include errors in the atmospheric model.

After correcting for systematic biases based on differences between observations and computations (BTDoc) in cloud-free conditions, the random error $\Delta T_{BG}$ in $T_{BG}$ is set from the standard deviation of the resulting distributions of BTDoc. Over ocean, nighttime and daytime standard deviations at 12.05 μm are similar, and found smaller than over land, where the deviations tend to be larger during daytime than at night. For simplicity, $\Delta T_{BG}$ at 12.05 μm is set to ± 1K over ocean, and to ± 3K over land for both night and day. Standard distributions of BTDoc(10) - BTDoc(12) indicate whether the errors in $T_{BG}$ at 10.6 and 12.05 μm are canceling out or not, after accounting for the contribution from the observations, which is estimated to √2x0.3 = 0.45 K. Standard deviations of [BTDoc(10) - BTDoc(12)] are found smaller than 0.5 K over ocean and over land during nighttime, which indicates that the errors in $T_{BG}$ at 12.05 μm and at 10.6 μm can be considered identical. They are found locally up to 1 K during daytime over land, which could reflect a variability of the 10-12 difference in surface emissivity, but also the presence of residual clouds. As a result, $\Delta T_{BG10}$ is assumed always equal to $\Delta T_{BG12}$.

Finally, the relative uncertainty $\Delta N/N$ is written as:

$$\left(\frac{\Delta N}{N}\right)^2 = \left[\frac{1}{f}\frac{\partial f}{\partial x}\cdot x\cdot\left(\frac{\partial\tau_{12}}{\tau_{12}\cdot\partial T_{BG}}-\frac{\partial\tau_{10}}{\tau_{10}\cdot\partial T_{BG}}\right)+\frac{\partial\tau_{12}}{\tau_{12}\cdot\partial T_{BG}}\right]^2\cdot\Delta T_{BG}^2 + \left[\frac{1}{f}\frac{\partial f}{\partial x}\cdot x\cdot\left(\frac{\partial\tau_{12}}{\tau_{12}\cdot\partial T_{BB}}-\frac{\partial\tau_{10}}{\tau_{10}\cdot\partial T_{BB}}\right)+\frac{\partial\tau_{12}}{\tau_{12}\cdot\partial T_{BB}}\right]^2\cdot\Delta T_{BB}^2$$

$$+\left[\left(\frac{1}{f}\frac{\partial f}{\partial x}\cdot x+1\right)\cdot\frac{\partial\tau_{12}}{\tau_{12}\cdot\partial T_{m12}}\right]^2\cdot\Delta T_{m12}^2 + \left[\left(\frac{1}{f}\frac{\partial f}{\partial x}\cdot x\right)\cdot\frac{\partial\tau_{10}}{\tau_{10}\cdot\partial T_{m10}}\right]^2\cdot\Delta T_{m10}^2 \qquad (A8)$$

with

$$\frac{1}{f}\frac{\partial f}{\partial x}\cdot x = \frac{2a2x^2+a1x}{a2x^2+a1x+a0} - \frac{2b2x^2+b1x}{b2x^2+b1x+b0} \qquad (A9)$$



**Acknowledgements**: This research was primarily supported by the Office of Science (BER), US Department of Energy and by the NASA CALIPSO project. Additional support was provided by NASA grant NNX16AM11G. Dr. Martina Krämer is gratefully acknowledged for providing us with the curve fits that describe her 2009 data set of cirrus cloud in situ data. CALIPSO products are available at the Atmospheric Science Data Center of the NASA

Langley Research Center and at the ICARE Data and Services Center in Lille (France).

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





Table 1. **Fraction of viable samples per season, over ocean and over land.**

| Fraction of viable samples - 2013 $\beta_{eff} > 1$ $\beta_{eff} > 1.035$ | DJF | MAM | JJA | SON |
|---|---|---|---|---|
| Sea | 0.94 0.85 | 0.93 0.84 | 0.92 0.82 | 0.93 0.83 |
| Land | 0.90 0.81 | 0.89 0.79 | 0.87 0.78 | 0.91 0.82 |



**Table 2. Sampled cirrus cloud frequency of occurrence for each 30-degree latitude zone, and also for the entire globe (last line) during 2008 and 2013.**

| Occurrence of selected conditions (%) during 2008 (Dec 2007 to Nov 2008) | | | | |
|---|---|---|---|---|
| | DJF | MAM | JJA | SON |
| 60N-82N | 0.49 | 0.2 | 0.21 | 0.22 |
| 30N-60N | 0.74 | 0.96 | 0.60 | 0.73 |
| 0N-30N | 1.58 | 1.90 | 2.02 | 1.70 |
| 30S-0S | 1.58 | 1.46 | 0.75 | 1.07 |
| 60S-30S | 0.25 | 0.39 | 0.47 | 0.36 |
| 82S-60S | 0.16 | 0.19 | 0.31 | 0.72 |
| Full globe | 0.81 | 0.86 | 0.73 | 0.80 |

| Occurrence of selected conditions (%) during 2013 (March 2013 to Feb 2014) | | | | |
|---|---|---|---|---|
| | DJF | MAM | JJA | SON |
| 60N-82N | 0.61 | 0.31 | 0.16 | 0.24 |
| 30N-60N | 0.90 | 0.98 | 0.56 | 0.65 |
| 0N-30N | 1.43 | 1.82 | 1.86 | 1.76 |
| 30S-0S | 1.58 | 1.47 | 0.79 | 1.05 |
| 60S-30S | 0.30 | 0.36 | 0.46 | 0.35 |
| 82S-60S | 0.09 | 0.20 | 0.42 | 0.61 |
| Full globe | 0.82 | 0.86 | 0.71 | 0.78 |





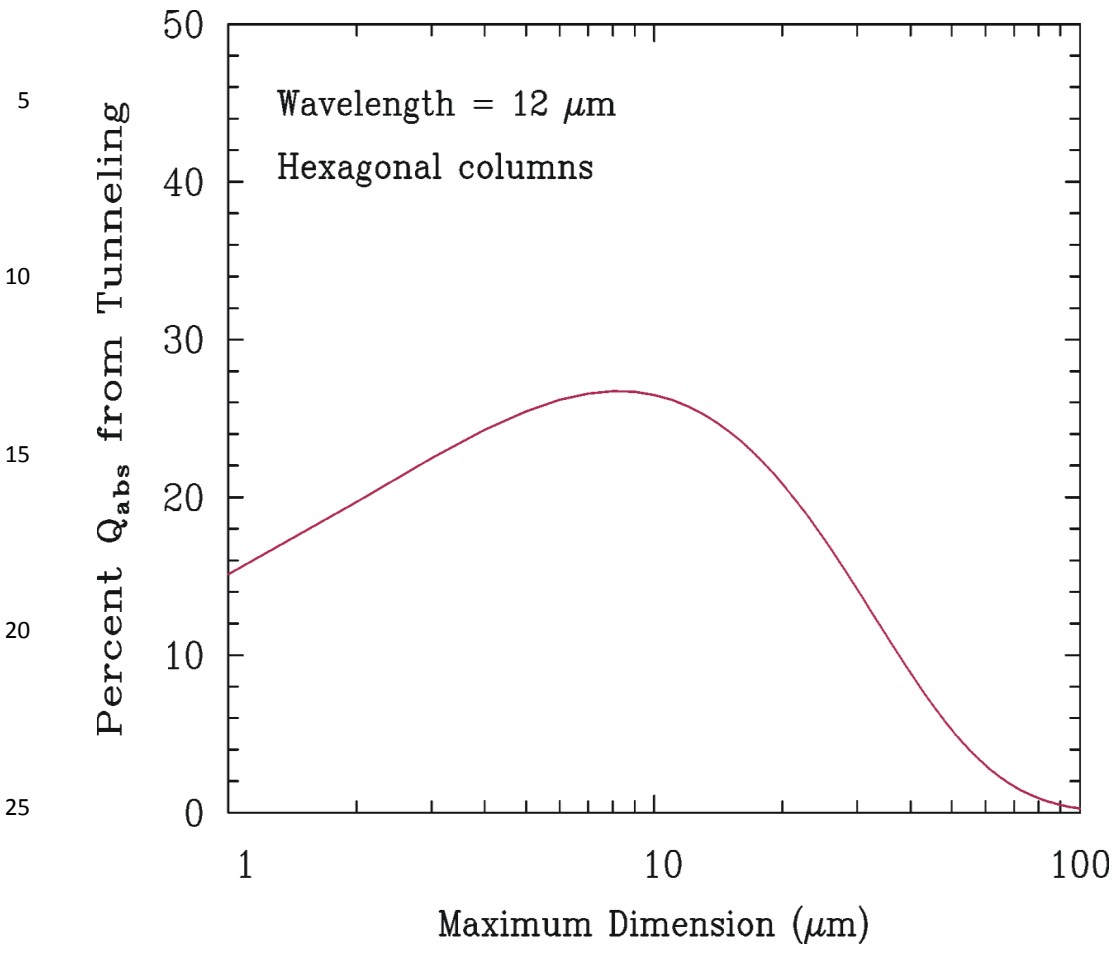

Figure 1: Percent contribution of wave resonance absorption to the overall absorption efficiency at 12 μm wavelength as a function of maximum dimension D for hexagonal columns, as estimated by the MADA.







**Figure 2:** Dependence of N/IWC on the effective absorption optical depth ratio $\beta_{eff}$ as predicted from the method of Parol et al. (1991), based on PSD from SPARTICUS and TC4. The curve-fit equation is given with variance ($R^2$) and root mean square error indicated.





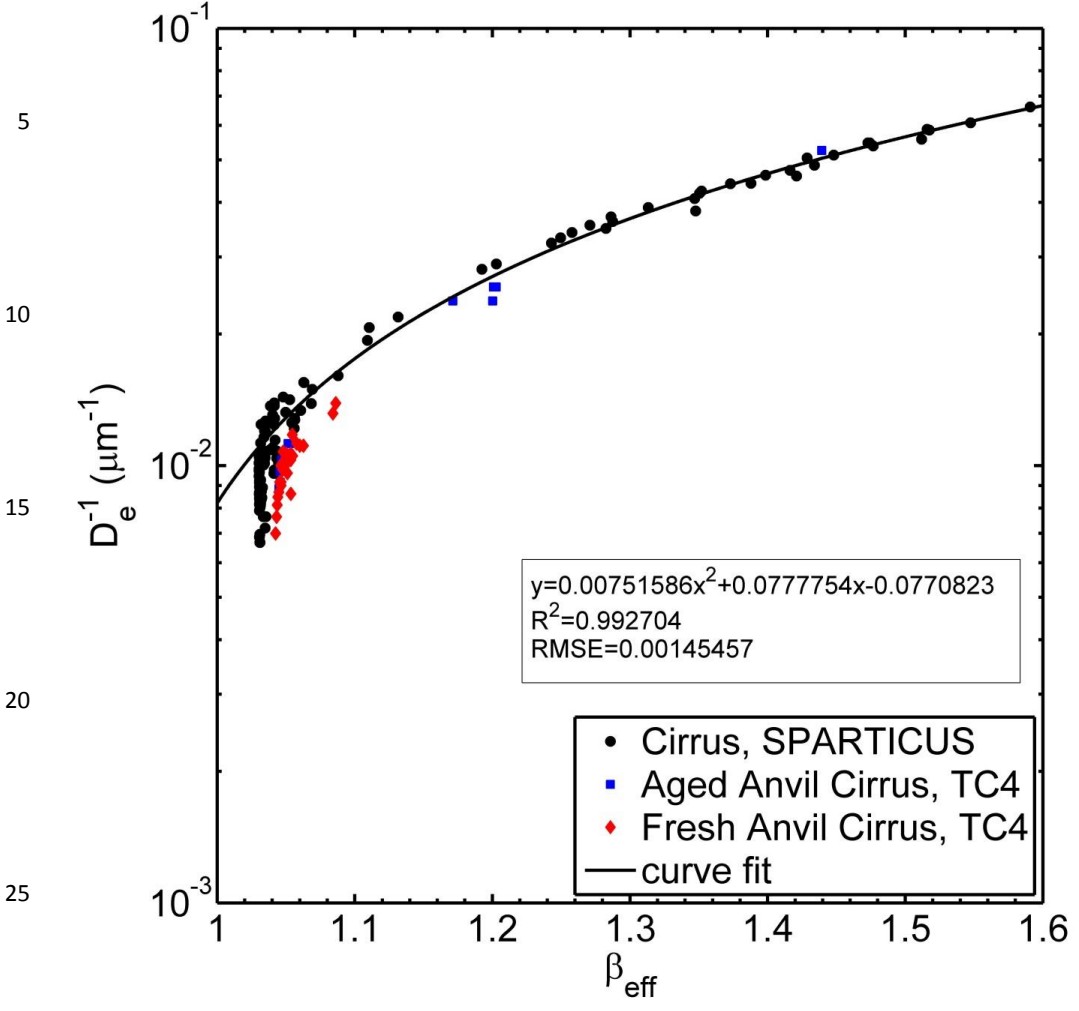

**Figure 3:** Dependence of the PSD effective diameter $D_e$ on the effective absorption optical depth ratio $\beta_{eff}$ as predicted from the method of Parol et al. (1991), based on PSD from SPARTICUS and TC4. The curve-fit equation is given with variance ($R^2$) and root mean square error indicated.





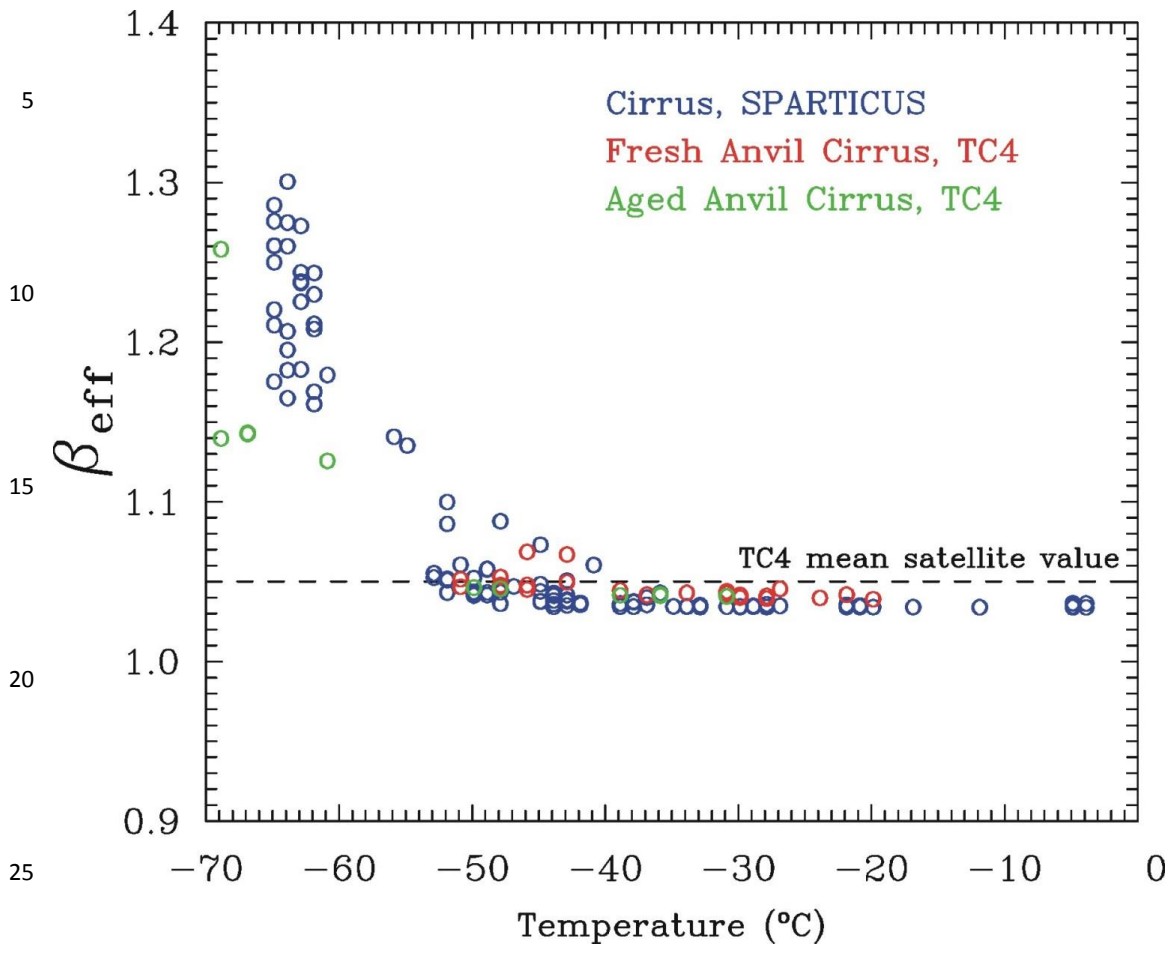

**Figure 4: Comparison of the mean value of MODIS retrievals of β$_{eff}$ during TC4 with β$_{eff}$ calculated from PSD obtained during TC4 and SPARTICUS. Note that the range of β$_{eff}$ has changed due to the MODIS channels used here.**







**Figure 5:** The $\beta_{eff}$ dependence of the term that converts $\tau_{abs}$ into visible optical depth in Eq. 4.



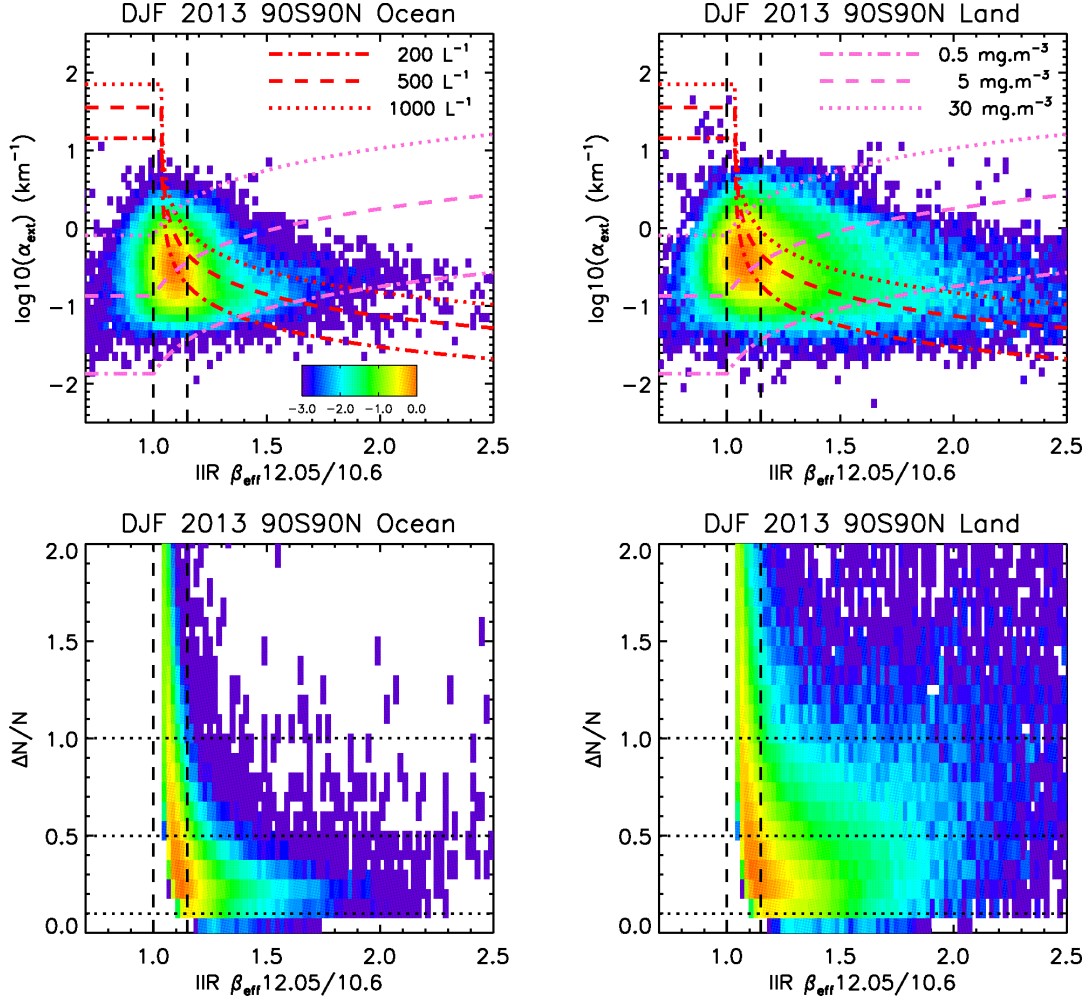

**Figure 6: Top:** The interrelationship between $\beta_{eff}$ (X-axis), layer extinction coefficient $\alpha_{ext}$ (km$^{-1}$)(Y-axis, log10 scale), IWC, and N. The red dashed lines are where N is equal to 200, 500, or 1000 L$^{-1}$. The pink dashed lines are where IWC is equal to 0.5, 5, or 30 mg.m$^{-3}$. **Bottom:** 2D-distribution of $\beta_{eff}$ (X-axis) and relative uncertainty estimate $\Delta N/N$. The color bar gives the log of number of samples normalized to the maximum value. Relative uncertainty tends to be considerably smaller at larger $\beta_{eff}$ values. Left: ocean; right: land; all latitudes; based on December 2013, January and February 2014.



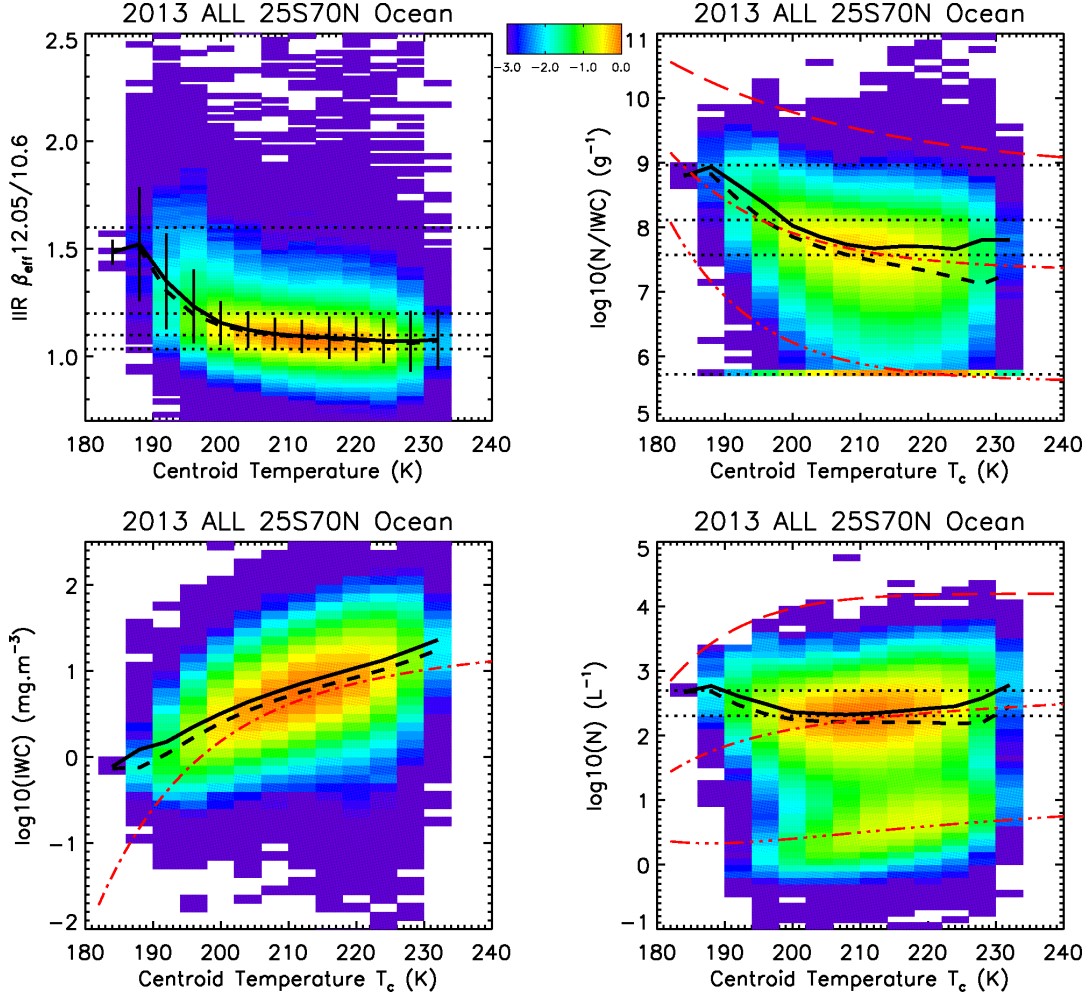

**Figure 7: Comparisons of the means (solid curves) and medians (black dashed) of retrieved N/IWC, IWC and N with corresponding in situ measurements from Krämer et al. (2009) shown by the red dashed curves; top and bottom being minimum and maximum values and middle red curve being the middle value. The original retrievals of $\beta_{eff}$ are shown in the upper left panel where the lower dotted line denotes 1.035. Retrievals are over the ocean and were averaged over all seasons and the indicated latitudes during 2013. $T_c$ is the representative cloud temperature. The dotted lines regarding log(N) vs. T indicate 200 $L^{-1}$ and 500 $L^{-1}$. Color code: number of samples were normalized to the maximum value on a decimal log scale.**



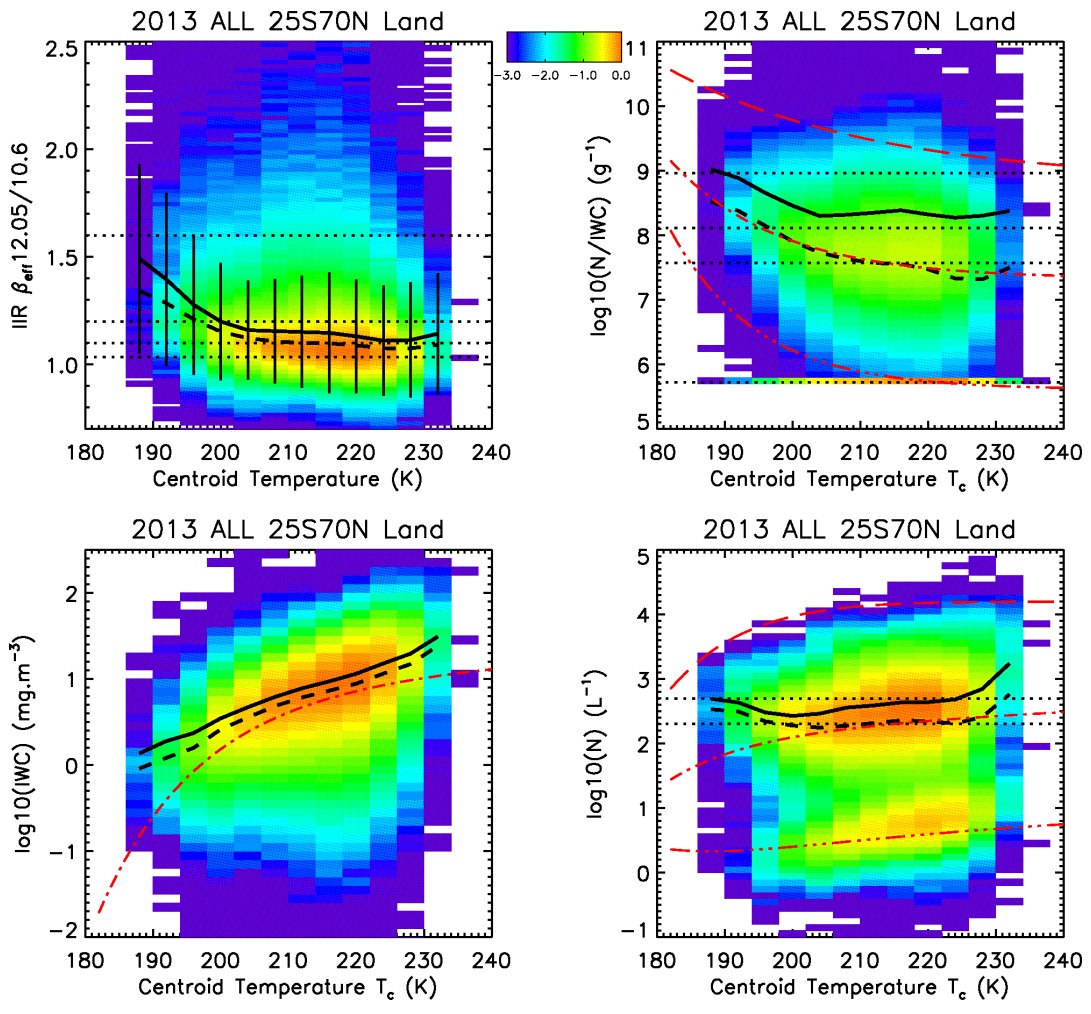

**Figure 8: Same as Fig. 7 except for retrievals over land.**





**Figure 9: Frequency of occurrence (indicated by legend in center) of sampled cirrus clouds for 2008 for each season.**





**Figure 10: Same as Fig. 9 but for 2013.**



5      **Figure 11:   Fraction of sampled cirrus clouds (indicated by legend in center) having N > 500 L$^{-1}$ for 2008 for each season.**







5 **Figure 12:** **Same as Fig. 11 except for 2013.**







**Figure 13:** Left column: Fraction of sampled cirrus with N > 500 L⁻¹ for each 2-degree latitude point (comprised of at least 15 samples) with uncertainties (±ΔN) for land (red) and ocean (blue) for each season during 2013. Right column: same as left column, but for fraction of sampled cirrus having N > 500 L⁻¹ and β_eff >1.15.





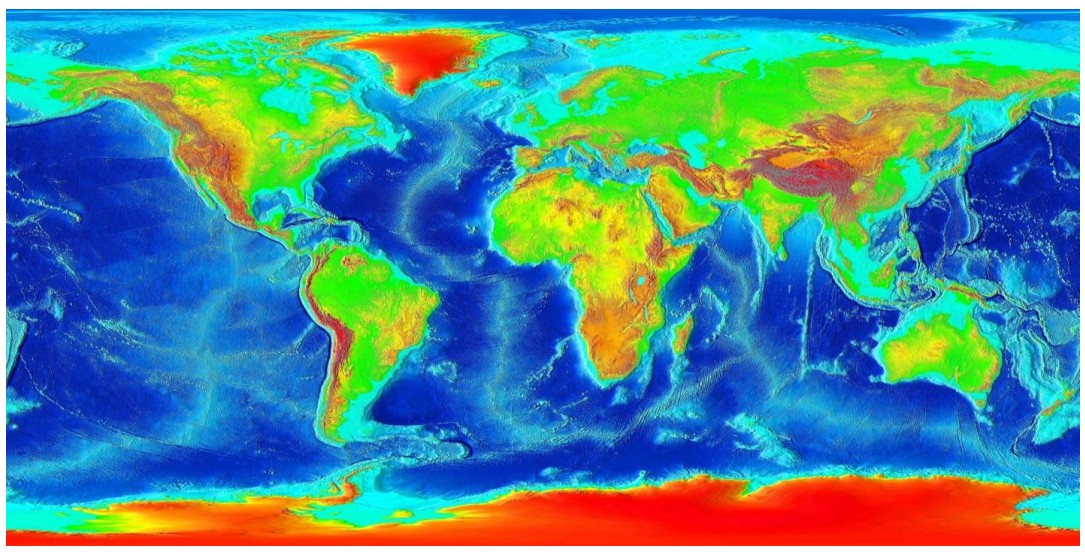

**Figure 14:  Elevation map for the Earth.  Over land, green is lowest and greyish-brown is highest.  Source: Wikipedia.**





**Figure 15:** Histograms for the number of DJF 1-km$^2$ selected cirrus cloud samples (black), the number of samples having N > 500 L$^{-1}$ (orange), the number having N > 500 L$^{-1}$ and $\beta_{eff}$ > 1.15 (red), and the number having $\beta_{eff}$ > 1.15 (blue) as a function of cloud centroid altitude. The fraction of these last 3 quantities relative to the number of selected samples is given under "Fraction" by the corresponding color. This is done for 6 latitude zones (rows) and for ocean only (left column), land only (middle column) and ocean + land (right column), and averaged for 2008 and 2013.





Figure 16: Same as Fig. 15 but for MAM.





Figure 17: Same as Fig. 15 but for JJA.



Figure 18: Same as Fig. 15 but for SON.



5    **Figure 19: Median value of retrieved effective diameter (Dₑ) for 2008 for each season over the temperature range of 206 K to 218 K. The color bar gives Dₑ in microns.**






5     **Figure 20: Same as Fig. 19 except for 2013.**





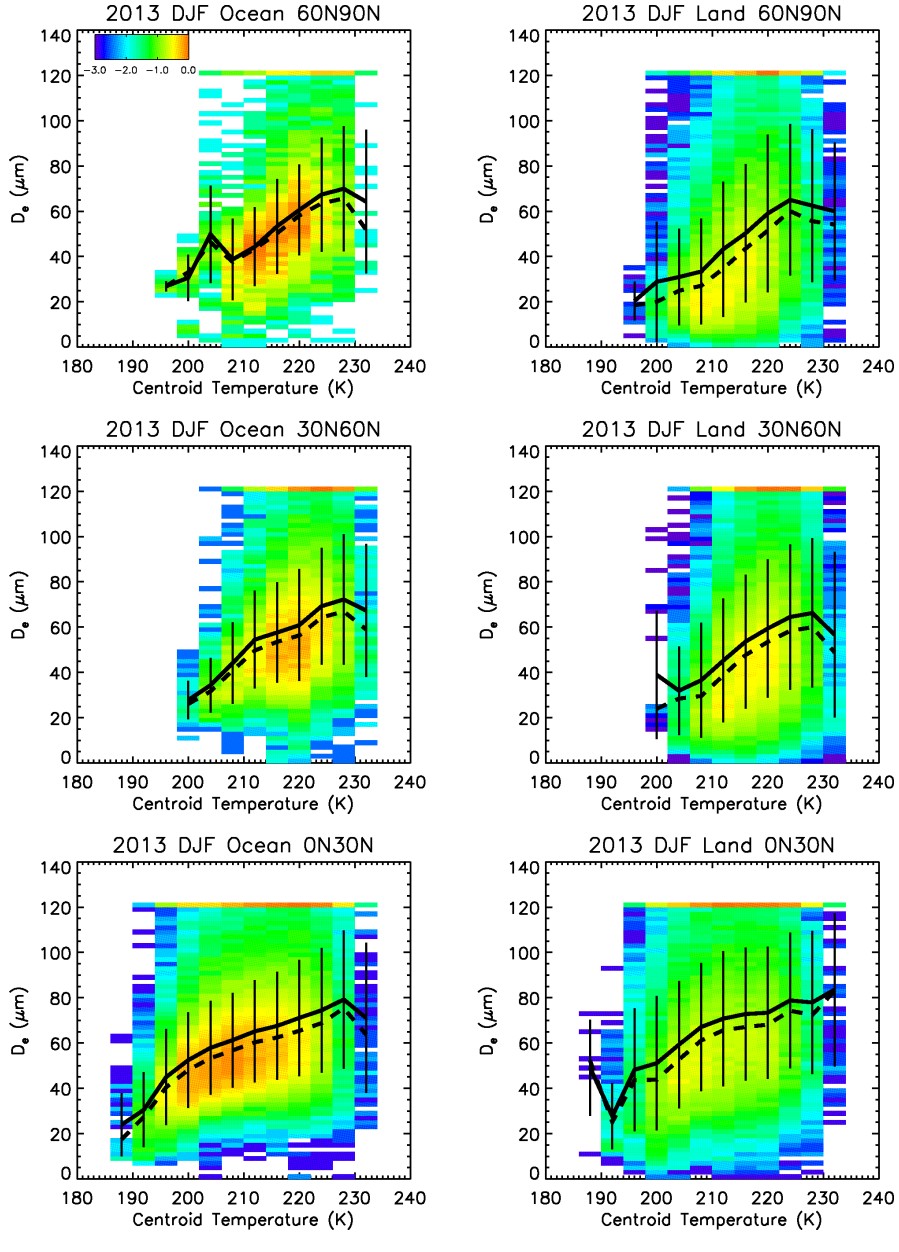

**Figure 21: Temperature dependence of the retrieved effective diameter ($D_e$ in microns) for three latitude zones in the Northern Hemisphere over both ocean and land during the winter of 2013-2014.**





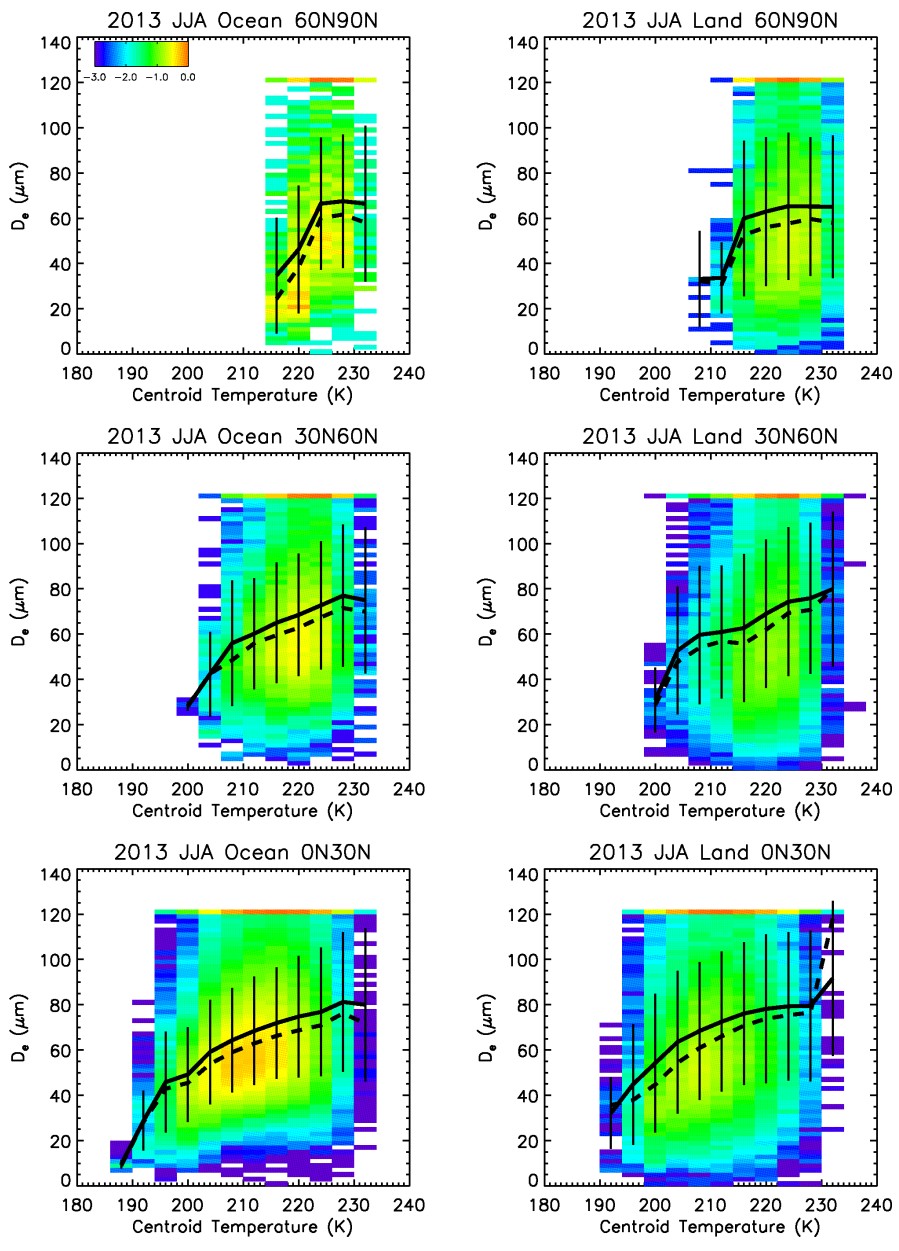

**Figure 22: Same as Fig. 21, but for the summer of 2013.**

