# Peer review of "CALIPSO observations of the dependence of homo- and heterogeneous ice nucleation in cirrus clouds on latitude, season and surface condition"

_Atmospheric Chemistry and Physics, 2016_

## Referee Comment (RC1) · Anonymous Referee #1 · 7 Jan 2017

**Review of "CALIPSO observations of the dependence of homo- and heterogeneous ice nucleation in cirrus clouds on latitude, season and surface condition" by D. Mitchell et al.**

This manuscript describes a new algorithm for retrieving cirrus ice concentration and effective radius from the CALIPSO satellite IIR and CALIOP measurements. The retrieved ice concentrations are used to derive regional and seasonal distributions of the fractions of cirrus formed by homogeneous and heterogeneous ice nucleation. If the retrievals are quantitatively accurate, the global distributions of cirrus ice concentrations and effective radii would be quite valuable in themselves, and the resulting conservative estimates of homogeneous freezing predominance would be a significant

discovery. However, I do not believe the manuscript is suitable for publication in its current form. I strongly suspect the retrieved ice concentrations far exceed those indicated by recent in situ observations. The comparison measurements published by Krämer et al. (2009) included in the manuscript does not provide a clear assessment of the retrieved quantities, and readily-available recent in situ measurements based on improved instrumentation should be used instead. If the authors can provide a clear comparison between the retrieved quantities and the recent in situ observations, and if the retrieved ice concentrations do agree well with these measurements, then the revised manuscript would be a valuable contribution to ACP.

Note that, given the possibility of a fundamental overestimation of ice concentrations from the retrieval, I have not provided detailed comments on the results shown beyond Figure 13 since these results are entirely dependent on the accuracy of the retrievals.

**General comments:**

1. There seem to be a number of approximations, corrections, and estimates involved in the retrieval procedure, and it is difficult to evaluate the potential errors and uncertainties associated with each of these steps. As a result, it is difficult to evaluate the uncertainties in retrieved ice concentrations used to distinguish between heterogeneous and homogeneous nucleation. Therefore, I think it is essential to include a proper comparison between the retrieved ice concentrations and in situ measurements of ice concentrations. The 2D-S measurements, which are most likely to be relatively free of shattering artifacts, would be ideal for this comparison. I suggest that the authors need to include comparisons between retrieved and in situ-measured frequency distributions of ice concentrations for the time periods and geographical regions corresponding to recent field experiments using the 2D-S instrument (SPARTICUS, MACPEX, TC4, AT-TREX, etc.). The in situ data can be subsetted to include appropriate extinction and temperature ranges for ideal comparison with the retrieved ice concentrations. My general impression is that the occurrence frequency of ice concentrations exceeding 500/L derived from the retrievals is far higher than indicated by in situ measurements, but a

proper comparison would clearly show whether such a discrepancy truly exists.

2. As shown by a number of recent modeling studies (e.g. Spichtinger and Gierens, 2009b, ACP; Jensen et al., 2012, JGR; Murphy, 2014, GRL), even though homogeneous freezing can produce high ice concentrations just after nucleation, sedimentation and entrainment tend to reduce ice concentrations as the clouds evolve, and even if the clouds are produced by homogeneous freezing, relatively low ice concentrations tend to dominate over the full depth and lifecycle of cirrus clouds. Therefore, the frequency of occurrence of ice concentrations exceeding 500/L is undoubtedly a vast underestimate of the fraction of clouds produced by homogeneous freezing. Some discussion of this issue should be included in the paper.

**Specific comments:**

**1. Abstract:** perhaps you could say something specific here about how the retrieval algorithm works?

**2. Page 2, line 12:** It would be clearer to say "such as an aqueous aerosol" than "such as a liquid drop" since you've already stated that the you're focusing on the temperature regime where liquid can't exist.

**3. Page 4, line 5:** It would be good to note that the ice concentration produced by hom is most sensitive to cooling rate.

**4. Page 4, first paragraph:** It would also be worth mentioning that het and hom can both contribute in air parcels that are cooling rapidly and/or have low IN concentrations. In this case, hom typically dominates the total ice concentration.

**5. Page 4, second paragraph:** Barahona and Nenes (2009) used a parcel model that did not include sedimentation. As shown by a number of recent studies (see general comment above), sedimentation can produce large regions below the level of nucleation with relatively low ice concentrations. In other words, sedimentation from a layer produced by hom with high ice concentrations can feed a deep layer with low ice

concentrations. Furthermore, modeling studies show that entrainment and differential sedimentation will tend to wash out the high ice concentrations produced hom, such that aged clouds originally produced by hom will have low ice concentrations and might look very similar to clouds produced by het.

**6. Page 6, lines 19–22:** Why don't you use MACPEX data in addition to SPARTICUS data (same instruments lots of cloud measurements)? Additional references describing the missions, flights, and data should be included?

**7. Page 6, line 26:** So, is the first 2D-S size bin included in this analysis?

**8. Page 6, line 28–29:** It would be helpful to have some specific information about the method used to derive $\beta_{eff}$ from the 2D-S PSDs.

**9. Page 7, lines 14–15:** Was satellite imagery used to distinguish TC-4 "fresh" (attached) from "aged" (detached) cirrus? It would be more appropriate to consider the time since detrainment from convective updrafts. In any case, there doesn't seem to be any distinction between the fresh and aged cases in the N/IWC vs $\beta_{eff}$ dependence shown in Figure 2.

**10. Page 7, line 27:** It looks like most of the TC4 data had N/IWC $< 10^7$/g. Does this imply the method described here cannot be used for most of the TC4 data? Perhaps these are just lumped into the het category based on relatively low N/IWC?

**11. Figure 3:** It would be helpful to show the $D_e$ values on the right axis corresponding to the $(D_e^{-1})$ on the left axis.

**12. Figure 4:** Do satellite- and in situ-derived retrievals of $\beta_{eff}$ agree for SPARTICUS cirrus colder than about -54 C, or is the method just not applicable at the colder temperatures? This would imply that the method is only applicable for a narrow temperature range between about 220 and 235 K. Were MODIS-retrieved $\beta_{eff}$ values for SPARTICUS similar to those in TC4?

**13. Page 7:** SPARTICUS included both convective and in situ cirrus. Are the regression curves shown in Figures 3–5 significantly different for these two types of cirrus?

**14. Page 9, line 9:** Presumably, the authors mean Figures 2 and 3 here instead of Figures 4 and 5.

**15. Page 12, lines 1–2:** A brief description for how the uncertainty in retrieved ice concentration should be included in the body of the paper.

**16. Page 12, lines 15-17:** I disagree with the authors' dismissal of the shattering problem for cirrus colder than 240 K. A number of papers have shown that ice crystal shattering is a major problem for FSSP-type instruments even in colder cirrus and even with the shroud removed whenever relatively large ice crystals are present (e.g., Mc-Farquhar et al., 2007, GRL; Jensen et al., 2009, ACP). The authors should compare their results with more recent measurements using the 2D-S instrument with post-processing based on inter-arrival times to remove shattering artifacts (e.g., from SPAR-TICUS, MACPEX, TC4, and ATTREX). An appropriate comparison would be to show the frequency distributions of ice concentrations from the retrievals with frequency distributions of in situ-measured ice concentrations for appropriate extinction and temperature ranges.

**17. Figures 7 and 8:** The comparison between in situ-observed and retrieved ice concentrations is obscured by showing six orders of magnitude, but it appears that the most common retrieved ice concentrations are generally considerably higher than those measured in situ, particularly over land. The discrepancy would likely be more glaring if the 2D-S measurements were used (see general comment above).

**18. Page 15, lines 3-5:** As noted here, the selection criteria used result in a small sample of the total cirrus present being included in the analysis. Perhaps some discussion of whether the sampling bias tends to select lower or higher ice-concentration cirrus could be included.

**19. Figure 13:** The retrievals indicate very high N > 500/L fractions. Over land, the

fractions typically exceed 50% in the extratropics. Published statistics of ice concentration from TC4 (Jensen et al., 2009, ACP) and SPARTICUS/MACPEX (Jensen et al., 2013, JGR) show that the 2D-S in situ-measured ice concentrations very rarely exceed 500/L.

---

## Referee Comment (RC2) · Anonymous Referee #2 · 27 Jan 2017

The authors propose a novel way, using combined lidar and spectral IR space observations, in synergy with relationships of physical variables obtained from airborne measurements, to obtain ice crystal number consentrations of single-layer cirrus clouds within a range of optical depth between 0.3 and 3. These results may be of great interest and relevance, yet at present they raise a number of very serious concerns, and the results presented are overinterpreted (see major concerns below). Hence we advise major revisions.

Major concerns: 1.a. The selection criteria (T<235K, OD between 0.3 and 3, radiative contrast with surface > 20K, single layer) lead to only 2% of sampled cirrus frequency map statistics appears quite noisy. Why did the authors only analyze 2 years of data,

when much more data are availabe?

1.b. The range of optical depths considered ranges from 0.3 to 3, so very different clouds are mixed together. It is necessary to distinguish between optically thinner and thicker cirrus.

2. The introduction clearly describes motivations related to in situ formed cirrus clouds. Yet it does not seem that the authors have a way to filter out anvil cirrus in their analysis. More generally, 'liquid origin' cirrus, whether originating from anvils or from warm conveyor belts, should be isolated.

3. Going from an estimate of the ice cystal number N to a conclusion on the nucleation is quite a step. Obtaining information on N is already very interesting. It would be better if the authors stick to results on N rather than try and push too far their interpretation. 4.a beta, which is related to De and N/IWC, is determined from 2 wavelengths. the IIR instrument has yet a third channel which should be more sensitive; why are not the 3 channels used, especially since you want to get the information on both parameters?

4.b you derive beta for OD down to 0.3; at this OD the contribution from the atmosphere is quite important which should introduce biases. Was the effect studied?

4.c The relation between cirrus properties and beta is obtained from in situ measurements from aircraft campaigns. How this applies to satellite observations which have a very different spatial footprint is not straightforward.

5. From 2 campaigns, and distinguishing fresh and aged cirrus in one, one sees that fresh anvil cirrus is beyond the sensitivity (beta < 1.1) and the other data which show indeed a spread over beta are incredibly close to the fit. It would be more convincing to have this fit made from several campaigns. Especially aged anvils are not really the focus of your analyses.

6. There are a number of assumptions that go into the derivation, e.g. the shape of the ice particles (hexagonal columns down to sizes where they are not truly relevant).

7. In polar locations, one has to be very careful with the underlying surface: the emissivity of ice, or sea ice (aging sea ice or fresh snow), has to be treated with special care. Was this taken into account for the retrieval?

8. There are many figures (22), many of them multi-panel figures (18 panels for each of the figures 15 to 18), adding up to more than 120 panels. This is too much. Surely it is the authors' responsibility to summarize and synthesize their work in a reasonnable number of figures.

9. There exist other retrievals of ice crystal number concentrations, using lidar - radar synergy, for example from DARDAR products (Delano and Hogan J. Geophys. Res. 2008 and 2010). How do your results compare to these ?

Minor points

p1, line 26: relevancy or relevance?

p2, lines 6-7: a reference justifying this assertion would be useful

p2, l14: Koop [et al], 2000

p2, l22: There is a recent study on gravity wave fluctuations and their implications on nucleation processes which provides an update and complement to the Haag et al (2003) reference: Jensen, E. J., et al. (2016), High-frequency gravity waves and homogeneous ice nucleation in tropical tropopause layer cirrus, Geophys. Res. Lett., 43, 6629–6635, doi:10.1002/2016GL069426.

p3, l4-6: 'they draw from a limited number of field campaigns... of het and hom cirrus.' This sentence is too critical of past field campaigns. Of course field campaigns have their limitations, and of course it is necessary to complement them with global observations, but these should be presented more as complementary, more positively.

p3, l16: I have not found the explanation in the text of the GMAO acronym.

p9, section 3.2: I found the presentation a little awkward: the full equation is given first

(equation (4)), then explanations for the different parts of the equation are provided. It is fine to stick with this, but I suggest below an alternative suggestion: - information on beta_eff allows to obtain N/IWC; hence to obtain N we wish to estimate IWC. This is given by (5) (a reference could be useful here, although this may seem standard). - now in this expression, it is needed to estimate the effective layer- mean extinction coefficient, and this is given in (6). - putting the pieces together, an expression for N can be obtained from CALIPSO IIR and CALIOP (for Delta z_{eq}): eq (4).

p11, l14: the OD range limits the conclusions of this study to visible cirrus; hence by construction the subvisible thin cirrus near the tropopause tropical layer are not covered.

p14, lines 26-27: why this choice of only two years, 2008 and 2013?

p15, l 28: It is inappropriate to include a figure which does not contain a color bar allowing to read the altitudes corresponding to the colors. Moreover, the figure contains more information than needed as the oceans' bathymetry is included. Finally, the purpose of the figure is only to recall the general location of Earth's major topographic features, and I believe most readers will be familiar enough with this (a detailed knowledge of topography is not required in the present study).

p16, l22: it is a bit surprising that het cirrus would be the most abundant over oceans, where ice nuclei concentrations may be expected to be weaker than over land. This is rather the signature that the intensities of updrafts matter more than the abundance of ice nuclei, is that the proper way to understand it,

p17: Figures 15-18: There are 72 panels in those figues. This is too much. I think the authors should identify patterns and show a more limited number of figures to display the main patterns, and add only those panels which contribute to interesting deviations from the general pattern. The authors should For instance, for 60S-30S, the four figures (15 to 18) show the same pattern, with variations that can be summarized in the text, or by a choice of fewer figures (just JJA and DJF). It is true that, with the figures available,

one could spot small variations in the details; but at this stage, given the uncertainties which one should keep in mind for a new, innovative method of remote-sensing, one should give too much weight to the details of the distributions.

p17, l27-28: the statement seems a bit exagerated, pushing the evidence further than should be.

p18, l1: the precision is unnecessary, 0.43 +/- 0.05 seems sufficient, given the uncertainties.. This is true in many other places of the text, e.g. p20, lines 2 and 6....

p18, l20: it is very good that this is recalled, this should be emphasized also in other parts of the text, e.g. in the conclusion.

p20, l9-10: clouds up to 25km: are these Polar Stratospheric Clouds? In which case they should be distinguished from cirrus. Their physics are not the same. It would not make sense to mix the two populations.

p20, Figues 19 and 20: the maps are so patchy! It would be better to average together 2008 and 2013, to analyze more data (why just two years?) and probably to average zonally.

p22, line 2: reverse order: 'exhibit a hom-het annual cycle similar to ...'

p22, l25: I am not sure that 'implies' is the proper word here. Should this rather be 'This accounts for our understanding that in situ cirrus...'

p23, l8-9: This sentence is a bit mysterious. In particular, what is meant by 'mechanistic'?

p23, l22: any ideas for the reason of these differences?

p24, l20: change to a comma: 'wave resonance absorption, a process...'

p24, l20-22: this point will be more convincing if more campaigns are used to establish the relationship.

[Figure]

p24, l25-29: this is an interesting point.

p25, l15: split into 2 sentences, : ...solar zenith andgle. With the SW...'

p26: these last paragraphs on Arctic Amplification are too long given the speculative nature of the arguments at this point.

---

## Author Comment (AC1) · 7 Apr 2017

Response to the referee comments on the manuscript:

Title: CALIPSO observations of the dependence of homo- and heterogeneous ice nucleation in cirrus clouds on latitude, season and surface condition Authors: David L. Mitchell, Anne Garnier, Melody Avery, and Ehsan Erfani Article reference: acp-2016-1062

We wish to thank the referees for their detailed and helpful comments on our paper. As you will see below we have responded to all of the comments with revisions designed to address the concerns of the referees. In the formatting below, the original referee

[Figure]

comments appear in black and our responses appear in blue and are labeled "Author response:"

Referee Comments:

Anonymous Referee #1:

This manuscript describes a new algorithm for retrieving cirrus ice concentration and effective radius from the CALIPSO satellite IIR and CALIOP measurements. The retrieved ice concentrations are used to derive regional and seasonal distributions of the fractions of cirrus formed by homogeneous and heterogeneous ice nucleation. If the retrievals are quantitatively accurate, the global distributions of cirrus ice concentrations and effective radii would be quite valuable in themselves, and the resulting conservative estimates of homogeneous freezing predominance would be a significant discovery. However, I do not believe the manuscript is suitable for publication in its current form. I strongly suspect the retrieved ice concentrations far exceed those indicated by recent in situ observations. The comparison measurements published by Krämer et al. (2009) included in the manuscript does not provide a clear assessment of the retrieved quantities, and readily-available recent in situ measurements based on improved instrumentation should be used instead. If the authors can provide a clear comparison between the retrieved quantities and the recent in situ observations, and if the retrieved ice concentrations do agree well with these measurements, then the revised manuscript would be a valuable contribution to ACP. Note that, given the possibility of a fundamental overestimation of ice concentrations from the retrieval, I have not provided detailed comments on the results shown beyond Figure 13 since these results are entirely dependent on the accuracy of the retrievals.

General comments:

1. There seem to be a number of approximations, corrections, and estimates involved in the retrieval procedure, and it is difficult to evaluate the potential errors and uncertainties associated with each of these steps. As a result, it is difficult to evaluate the uncertainties in retrieved ice concentrations used to distinguish between heterogeneous and homogeneous nucleation. Therefore, I think it is essential to include a proper comparison between the retrieved ice concentrations and in situ measurements of ice concentrations. The 2D-S measurements, which are most likely to be relatively free of shattering artifacts, would be ideal for this comparison. I suggest that the authors need to include comparisons between retrieved and in situ-measured frequency distributions of ice concentrations for the time periods and geographical regions corresponding to recent field experiments using the 2D-S instrument (SPARTICUS, MACPEX, TC4, AT-TREX, etc.). The in situ data can be subsetted to include appropriate extinction and temperature ranges for ideal comparison with the retrieved ice concentrations. My general impression is that the occurrence frequency of ice concentrations exceeding 500/L derived from the retrievals is far higher than indicated by in situ measurements, but a proper comparison would clearly show whether such a discrepancy truly exists.

Author response: We recognize the need for additional comparisons with cirrus cloud in situ data, using reliable probes like the 2D-S, and plan to do this in a future paper. We also feel that some of this can be included in this paper (which is already rather long), and thus have added a new section (Sect. 4.3.2; Comparisons with SPARTICUS data). Section 4.3.2 begins with a brief discussion of how the 2D-S probe design and post-processing of the probe data greatly reduces shattering artefacts, and how the SPARTICUS campaign was conceived to produce a statistically meaningful cirrus cloud in situ dataset that was relatively free from shattering artefacts. We then present comparisons between retrieved and in situ ice particle concentration ($N$) measurements from SPARTICUS, binning both the $N$ retrievals and $N$ measurements by temperature (5 °C intervals). Since the main concern is whether the fraction of $N > 500/L$ from the retrieval and the in situ measurements are consistent, we present the retrieved and in situ $N$ measurements in that format in Fig. 9 (i.e. as the fraction of $N > 500/L$). Since $N$ in cirrus clouds is highly variable in nature, perfect agreement on a bin-by-bin basis is not expected, but overall there is consistency between the in situ measurements and the retrievals. Also important is that when temperature $T > 235$ K (the threshold for

homogeneous ice nucleation; henceforth hom), the N fraction for both threshold values (N > 500/L and N > 250/L) is zero or near zero, respectively, which should be the case for a hom threshold. More details are given in Sect. 4.3.2.

2. As shown by a number of recent modeling studies (e.g. Spichtinger and Gierens, 2009b, ACP; Jensen et al., 2012, JGR; Murphy, 2014, GRL), even though homogeneous freezing can produce high ice concentrations just after nucleation, sedimentation and entrainment tend to reduce ice concentrations as the clouds evolve, and even if the clouds are produced by homogeneous freezing, relatively low ice concentrations tend to dominate over the full depth and lifecycle of cirrus clouds. Therefore, the frequency of occurrence of ice concentrations exceeding 500/L is undoubtedly a vast underestimate of the fraction of clouds produced by homogeneous freezing. Some discussion of this issue should be included in the paper.

Author response: The above is a prediction from cloud models that are not always representative of actual cirrus clouds. While these processes (sedimentation and entrainment) certainly are active, their impact on N is not a closed issue. Rather than digress into modeling arguments, we have compared our results with SPARTICUS data in Fig. 9. We also show in Fig. 9 that a lower N threshold for hom of 250/L also appears valid based on SPARTICUS data (anything lower than 250/L was not a good threshold due to substantial N fractions for T > 235 K). These results support the referees claim that N > 500/L is a conservative threshold, but only by a factor of about 2. Furthermore, we acknowledge that sedimentation and entrainment will dilute N relative to nucleation onset on p. 4, lines 26-28, where it states: "This threshold is especially conservative when one considers that entrainment and ice sedimentation effects are predicted to dilute the N in cirrus formed through large-scale ascent (Spichtinger and Gierens, 2009; Jensen et al., 2012b)."

Specific comments: 1. Abstract: perhaps you could say something specific here about how the retrieval algorithm works?

[Figure]

Author response: Good idea. A sentence in the abstract has been modified to read as follows: "A new satellite remote sensing method is described whereby the sensitivity of wave resonance absorption to small ice crystals is exploited in this study to estimate cirrus cloud ice particle number concentration and the relative contribution of hom and het to cirrus cloud formation as a function of altitude, latitude, season and surface type (e.g. land vs. ocean)."

2. Page 2, line 12: It would be clearer to say "such as an aqueous aerosol" than "such as a liquid drop" since you've already stated that the you're focusing on the temperature regime where liquid can't exist.

Author response: Agreed. Change has been made exactly as suggested.

3. Page 4, line 5: It would be good to note that the ice concentration produced by hom is most sensitive to cooling rate.

Author response: This has already been mentioned on p. 2, lines 20-22, and on p. 4, lines 4, 6, and 7 of the original ACPD paper.

4. Page 4, first paragraph: It would also be worth mentioning that het and hom can both contribute in air parcels that are cooling rapidly and/or have low IN concentrations. In this case, hom typically dominates the total ice concentration.

Author response: Agreed. New text has been added on page 4, lines 17-19: "Given these criteria for het and hom, one can see how both het and hom can contribute to N in a strong cirrus cloud updraft, with het producing the first crystals, but due to insufficient ice surface area to prevent the RHi from climbing, the hom threshold can be reached with crystals produced by hom usually dominating N."

5. Page 4, second paragraph: Barahona and Nenes (2009) used a parcel model that did not include sedimentation. As shown by a number of recent studies (see general comment above), sedimentation can produce large regions below the level of nucleation with relatively low ice concentrations. In other words, sedimentation from a

layer produced by hom with high ice concentrations can feed a deep layer with low ice concentrations. Furthermore, modeling studies show that entrainment and differential sedimentation will tend to wash out the high ice concentrations produced hom, such that aged clouds originally produced by hom will have low ice concentrations and might look very similar to clouds produced by het.

Author response: The Jensen et al. (2012, JGR) study was based on TTL cirrus, where the cirrus are long-lived near the tropopause forming through slow ascension. The cirrus studied by Spichtinger and Gierens (2009) were formed by synoptic-scale updrafts ranging from 0.05 m s-1 (standard run) to 0.10 m s-1. (We were not able to find the Murphy reference in GRL). For these cirrus, these modeling results make sense. But the cirrus having high N based on our retrievals are associated with mountain-induced waves that appear to form as the wave crests. The relatively high updrafts can increase the relative humidity with respect to ice (RHi) to produce "hom cirrus" (i.e. hom contributes to N), resulting in high N throughout the moist layer. Indeed, the Spichtinger and Gierens study also showed much higher N values throughout the cirrus cloud when w was increased from 0.06 to 0.08, and then to 0.10 m s-1. The CALIPSO results suggest that these wave cirrus can be advected for hundreds of km downwind. It is less clear how entrainment and sedimentation will affect N in these cirrus. Nonetheless, a sentence concerning these processes was added to this paragraph on p. 4, lines 26-28: "This threshold is especially conservative when one considers that entrainment and ice sedimentation effects are predicted to dilute the N in cirrus formed through large-scale ascent (Jensen et al., 2012b; Spichtinger and Gierens, 2009)."

6. Page 6, lines 19–22: Why don't you use MACPEX data in addition to SPARTICUS data (same instruments lots of cloud measurements)? Additional references describing the missions, flights, and data should be included?

Author response: This would make a rather long paper even longer. We have already begun to compare the CALIPSO retrievals with in situ data from other field campaigns, and plan a separate paper for these comparisons.

7. Page 6, line 26: So, is the first 2D-S size bin included in this analysis?

Author response: Yes, and new text has been added on p. 6, lines 30-31: "The data in the smallest size bin (5–15 $\mu$m) has greater uncertainty (Jensen et al., 2013a) but was used in this analysis since to exclude it would likely result in greater N error than the N uncertainty introduces."

8. Page 6, line 28–29: It would be helpful to have some specific information about the method used to derive $\beta$eff from the 2D-S PSDs.

Author response: This specific information is given in the same paragraph, continued on the next page, including Equations (2) and (3).

9. Page 7, lines 14–15: Was satellite imagery used to distinguish TC-4 "fresh" (attached) from "aged" (detached) cirrus? It would be more appropriate to consider the time since detrainment from convective updrafts. In any case, there doesn't seem to be any distinction between the fresh and aged cases in the N/IWC vs $\beta$eff dependence shown in Figure 2.

Author response: Since the N/IWC vs $\beta$eff dependence is similar for fresh and aged anvils, a digression into the methods used to determine how they were classified does not seem warranted. The small differences are believed due to PSD shape differences as shown in Mitchell et al. (2011; ref. given in paper). The criteria that SPEC, Inc. used to discriminate between fresh and aged anvils was whether the anvil was attached to the convective column or not (also mentioned in paper). For more information about the TC4 data, on p. 6 the reader is referred to Mitchell et al. (2011).

10. Page 7, line 27: It looks like most of the TC4 data had N/IWC < 107 /g. Does this imply the method described here cannot be used for most of the TC4 data? Perhaps these are just lumped into the het category based on relatively low N/IWC?

Author response: Yes, much of the TC4 data was collected between -20 and -40 °C where PSD tend to be relatively broad with larger mean sizes and median mass dimensions. For such PSD, $\beta$eff tends to be < 1.06 and N/IWC < 107/g, and since T < -38 °C for hom, ice produced between -20 and -38 °C must be from het. But since our method targets cirrus clouds with cloud base temperature < -38 °C, PSD at these temperatures tend to be relatively narrow and appropriate for sampling using our retrieval method. New text has been added on p. 8, lines 4-7: "Much of the data shown in Fig. 2 were derived from ice cloud PSD at temperatures between -20 and -38 °C where PSD tend to be relatively broad with low $\beta$eff values (< 1.06). But since this method targets cirrus clouds with cloud base temperature less than -38 °C, PSD at these temperatures tend to be relatively narrow and appropriate for sampling using this retrieval method."

11. Figure 3: It would be helpful to show the De values on the right axis corresponding to the (De−1) on the left axis.

Author response: Done.

12. Figure 4: Do satellite- and in situ-derived retrievals of $\beta$eff agree for SPARTICUS cirrus colder than about -54 C, or is the method just not applicable at the colder temperatures? This would imply that the method is only applicable for a narrow temperature range between about 220 and 235 K. Were MODIS-retrieved $\beta$eff values for SPARTICUS similar to those in TC4?

Author response: The purpose of Fig. 4 was to demonstrate the equivalency between $\beta$eff calculated from PSD and corresponding retrieved $\beta$eff, using the TC4 campaign to do this. Inclusion of SPARTICUS PSD appears to have confused matters, so the SPARTICUS PSD have been removed from Fig. 4. We have not obtained any MODIS-retrieved $\beta$eff values for SPARTICUS. New text has been added near the end of the paragraph addressing Fig. 4 on p. 8, lines 29-30: "$\beta$eff tends to be constant in Fig. 4 since the contribution of small (D < 50 $\mu$m) particles to N is relatively low for TC4 cirrus (Mitchell et al., 2011) and the MODIS channels used produce lower $\beta$eff values." This is consistent with our retrieval results that show the lowest N-fractions (for N > 500/L) occur over the tropics.

13. Page 7: SPARTICUS included both convective and in situ cirrus. Are the regression curves shown in Figures 3–5 significantly different for these two types of cirrus?

Author response: The regression curves in Figs. 2, 3 and 5 are based on SPARTICUS PSD data from both anvil cirrus and synoptic cirrus. The conformity of the SPARTICUS data with the regression curves shows that the anvil and synoptic PSD behave similarly. New text has been added to p. 7, lines 17-18: "Anvil cirrus data from SPARTICUS are also included in Fig. 2."

14. Page 9, line 9: Presumably, the authors mean Figures 2 and 3 here instead of Figures 4 and 5.

Author response: Yes; thank you for pointing this out!

15. Page 12, lines 1–2: A brief description for how the uncertainty in retrieved ice concentration should be included in the body of the paper.

Author response: Section 4.2 on retrieval uncertainties has been greatly expanded to accommodate this request. Most of the text is new.

16. Page 12, lines 15-17: I disagree with the authors' dismissal of the shattering problem for cirrus colder than 240 K. A number of papers have shown that ice crystal shattering is a major problem for FSSP-type instruments even in colder cirrus and even with the shroud removed whenever relatively large ice crystals are present (e.g., Mc-Farquhar et al., 2007, GRL; Jensen et al., 2009, ACP). The authors should compare their results with more recent measurements using the 2D-S instrument with post-processing based on inter-arrival times to remove shattering artifacts (e.g., from SPAR-TICUS, MACPEX, TC4, and ATTREX). An appropriate comparison would be to show the frequency distributions of ice concentrations from the retrievals with frequency distributions of in situ-measured ice concentrations for appropriate extinction and temperature ranges.

Author response: As noted, a new section has been added; Sect. 4.3.2. The first paragraph of this section addresses the problem of shattering and what has been done to minimize shattering. The rest of this section describes how 2D-S probe measurements from SPARTICUS (with post-processing based on inter-arrival times to remove shattering artifacts) have been used to test our retrievals. As recommended, comparisons show the frequency distributions of ice concentrations (expressed as N-fraction ratios) from the retrievals with the same frequency distributions of in situ-measured N-fractions as a function of temperature.

17. Figures 7 and 8: The comparison between in situ-observed and retrieved ice concentrations is obscured by showing six orders of magnitude, but it appears that the most common retrieved ice concentrations are generally considerably higher than those measured in situ, particularly over land. The discrepancy would likely be more glaring if the 2D-S measurements were used (see general comment above).

Author response: Figures 7 and 8 clearly show that for T > 200 K, there is good agreement between the median N values of the retrievals and the in situ measurements, with the in situ N median often being higher over ocean (and with improved agreement over land). As requested, 2D-S measurements are compared with the retrievals in Fig. 9.

18. Page 15, lines 3-5: As noted here, the selection criteria used result in a small sample of the total cirrus present being included in the analysis. Perhaps some discussion of whether the sampling bias tends to select lower or higher ice-concentration cirrus could be included.

Author response: In an attempt to (partially) address this comment, Section 4.1 describing the relationship between ïĄćeff, ïĄąext, IWC, and N has been developed. In particular, we write on p. 12, lines 11-12: "When $\alpha$ext is smaller than 0.05 km-1, as could be encountered in the case of OD smaller than 0.3 (removed from the cloud sampling), finding N larger than 500 L-1 is very unlikely". However, the range of possible ïĄćeff and ïĄąext in the case of opaque clouds is such that no systematic bias with respect to our cloud selection criteria can be anticipated. Overall, considering our selection criteria, we find no unambiguous evidence indicating a systematic bias towards either lower or higher N.

19. Figure 13: The retrievals indicate very high N > 500/L fractions. Over land, the fractions typically exceed 50% in the extratropics. Published statistics of ice concentration from TC4 (Jensen et al., 2009, ACP) and SPARTICUS/MACPEX (Jensen et al., 2013, JGR) show that the 2D-S in situ-measured ice concentrations very rarely exceed 500/L.

Author response: We agree that for TC4, the 2D-S in situ-measured ice concentrations very rarely exceed 500/L. Our CALIPSO retrievals also show this for the TC4 region and for the tropics in general where anvil cirrus dominates. One explanation given in our paper is that anvil cirrus are liquid origin cirrus, which, by definition, contain pre-existing ice, and this pre-existing ice provides a relatively large amount of ice surface area that would otherwise not be present at the time of ice nucleation. This will suppress the RHi, preventing hom from activating and preventing N from exceeding $\sim$ 500/L (or more commonly $\sim$ 250 L-1). A large portion of MACPEX cirrus were anvil cirrus and Fig. 1 in Jensen et al. (2013) shows that MACPEX cirrus sampling occurred in regions that were generally far from the Rocky Mtns. relative to SPARTICUS. We have added text on p. 19, lines 13-17: "Figures 12 and 13 show that relatively high N is found over the Rocky Mountains and downwind of the Rockies during winter-spring, and in Fig. 1 of Jensen et al. (2013), it is seen that most of the flights during SPARTICUS were flown in this same region (during winter-spring). Based on our retrievals, one would expect relatively high N during SPARTICUS relative to field campaigns less associated with high mountains, such as MACPEX (Jensen et al., 2013a, Fig 1). Figure 9 confirms that relatively high N was measured by aircraft during SPARTICUS." From January through April, only synoptic cirrus clouds were sampled during SPARTICUS, which is consistent with our findings in Fig. 9.

Muhlbauer et al. (2014, JGR) is another study devoted to SPARTICUS data analysis, and Fig. 6 of Muhlbauer et al. shows that the sampling frequency for N $\geq$ 500 L-1 is

substantial (» 1%) for synoptic cirrus whereas this fraction is quite low in the case of anvil cirrus.

Anonymous Referee #2

The authors propose a novel way, using combined lidar and spectral IR space observations, in synergy with relationships of physical variables obtained from airborne measurements, to obtain ice crystal number consentrations of single-layer cirrus clouds within a range of optical depth between 0.3 and 3. These results may be of great interest and relevance, yet at present they raise a number of very serious concerns, and the results presented are overinterpreted (see major concerns below). Hence we advise major revisions.

Major concerns:

1.a. The selection criteria (T<235K, OD between 0.3 and 3, radiative contrast with surface > 20K, single layer) lead to only 2% of sampled cirrus frequency map statistics appears quite noisy. Why did the authors only analyze 2 years of data, when much more data are available?

Author response: A future publication will provide more years of CALIPSO analysis using this retrieval, including trend analysis of the CALIPSO data record. The purpose of this paper is to introduce the retrieval method, compare it with some in situ measurements, and present two years of results that is sufficient for identifying cirrus cloud seasonal cycles and other characteristics. We have looked at three other years (2007, 2014, 2015) not reported in the paper, but the analysis was not as thorough as 2008 and 2013. This was done to confirm that the seasonal trends observed for 2008 and 2013 were found in these other years. To keep the paper's length reasonable, we have restricted the analysis to two years. Text has been added on p. 18, lines 10-11: "Some limited analysis was done for years 2007, 2014 and 2015 (not shown) to confirm that

the results presented here are found for other years as well."

1.b. The range of optical depths considered ranges from 0.3 to 3, so very different clouds are mixed together. It is necessary to distinguish between optically thinner and thicker cirrus.

Author response: The sampling statistics over many regions for a given season would not be acceptable if we subdivided the data into narrower optical depth (OD) intervals. An excellent paper by Hong and Liu (2015, J. Climate) presents a CloudSat-CALIPSO (DARDAR) global analysis of ice clouds (T < 0 °C) where the data is subdivided into various OD intervals like the referee suggests. However, the OD retrieval range in Hong and Liu is much greater than ours, and one of their OD categories ranges from 0.3 to 3.0 (the same as our retrieval range). Evidently, they felt that this OD range was appropriate for defining a cirrus cloud category.

2. The introduction clearly describes motivations related to in situ formed cirrus clouds. Yet it does not seem that the authors have a way to filter out anvil cirrus in their analysis. More generally, 'liquid origin' cirrus, whether originating from anvils or from warm conveyor belts, should be isolated.

Author response: Indeed it would be nice to remotely distinguish between liquid origin and in situ cirrus clouds, which can be done in field campaigns. Perhaps that will be done someday, but that is not the purpose of this paper, and it is beyond the scope of this paper.

3. Going from an estimate of the ice cystal number N to a conclusion on the nucleation is quite a step. Obtaining information on N is already very interesting. It would be better if the authors stick to results on N rather than try and push too far their interpretation.

Author response: Previously we were using the parcel model results from Barahona and Nenes (2009, ACP) to discriminate between hom and het, but now we have added a new section, Sect. 4.3.2 (Comparisons with SPARTICUS data) that provides additional justification for using N to discriminate between hom and het. The Barahona and Nenes study predicted the het-to-hom transition begins around 200 L-1 while the SPARTICUS results suggest this transition occurs (for this dataset) at about 250 L-1, based on N-fractions at temperatures below and above -40 °C.

4.a beta, which is related to De and N/IWC, is determined from 2 wavelengths. the IIR instrument has yet a third channel which should be more sensitive; why are not the 3 channels used, especially since you want to get the information on both parameters?

Author response: Firstly, the Imaging Infrared Radiometer (IIR) has channels at 8.7, 10.6, and 12.05 $\mu$m. The answer to this question lies in Fig. 1 of Mitchell et al. (2010, JAS), where it is seen that the real refractive index, mr, at 8.7 $\mu$m and 12.05 $\mu$m wavelength are very similar (near 1.3). This retrieval uses absorption produced through wave resonance (i.e. photon tunneling) to retrieve N, De, etc. since wave resonance absorption is very sensitive to small ice crystals. For ice, wave resonance absorption is proportional to the real refractive index (see Fig. 4 in Mitchell, 2000, JAS). The retrieval principle used requires two wavelengths that have very different contributions from wave resonance absorption by ice so that one IIR channel receives emissions enriched by wave resonance absorption while the other channel receives emissions having very little contribution from wave resonance absorption. Since mr is comparable at 8.7 $\mu$m and 12.05 $\mu$m wavelengths, these two channels are not suitable for retrieving N. But the 10.6 $\mu$m channel is near the mr minimum, making it very suitable for retrieving N. Using the 8.7 $\mu$m with the 10.6 $\mu$m channel is not suitable either since the imaginary refractive index mi at 8.7 $\mu$m is much lower than at 12.05 $\mu$m, giving cloud emission a strong dependence on both Beer's law type absorption and wave resonance absorption rather than primarily wave resonance absorption (as it is at 12.05 $\mu$m). It is feasible to retrieve De using the 8.7 $\mu$m channel (e.g. Dubuisson et al., 2008, JAMC; Garnier et al., 2013, JAMC), but by using the same channels for both De and N, the two retrievals are self-consistent with respect to PSD moments. In addition, the analytical PSD calculations for $\beta$eff are more accurate when using the 10.6/12 combination relative to the 8.7/12 combination (Figs. 1b and 2b in Garnier et al., 2013, JAMC) since there is less multiple scattering with the 10.6/12 combination. New text has been added on p. 8, lines 16-20: "Although it is feasible to retrieve De using the IIR 8.7 $\mu$m channel (e.g. Dubuisson et al., 2008; Garnier et al., 2013), by using the same channels for both De and N, the two retrievals are self-consistent with respect to PSD moments. In addition, the analytical PSD calculations for $\beta$eff are more accurate when using the 10.6/12 combination relative to the 8.7/12 combination (Figs. 1b and 2b in Garnier et al., 2013, JAMC) since there is less multiple scattering with the 10.6/12 combination."

4.b you derive beta for OD down to 0.3; at this OD the contribution from the atmosphere is quite important which should introduce biases. Was the effect studied?

Author response: the contributions from the atmosphere and from the surface are taken into account in the computation of the effective emissivity and ultimately of the absorption optical depth. This contribution is the clear sky background radiance RBG in Eq. 8. As described on p.10, the accuracy of the computed RBG, expressed in terms of brightness temperature TBG for more convenience, has been evaluated by analyzing differences between observed and computed brightness temperatures in clear sky conditions. For this study, computed TBG have been corrected for remaining biases for both IIR channels. Our retrievals are applied only when the cloud layer of interest is the only cloud in the column (see beginning of Sect. 4) to ensure that the background radiance is indeed a "clear sky" radiance. The uncertainty in TBG is estimated to be the standard deviation of the distributions of differences between observed and computed TBG after correction for the initial biases. This was detailed in the appendix, but is now explained also in Sect. 4.2 for more clarity after a comment by referee #1. The uncertainty in TBG is $\pm$ 1K over ocean and $\pm$ 3K over land, and is one of the contributors to the uncertainty in N (see Eq. A8 in the appendix).

4.c The relation between cirrus properties and beta is obtained from in situ measurements from aircraft campaigns. How this applies to satellite observations which have a

very different spatial footprint is not straightforward.

Author response: This is always an uncertainty with remote sensing since one does not know how representative the in situ measurements are when applied to the entire planet. New text has been added on p. 8, lines 20-22: "The use of these relationships (i.e. Figs. 2 and 3) in satellite remote sensing necessarily assumes that the in situ measurements they are based on are globally representative; an assumption that is difficult to test."

5. From 2 campaigns, and distinguishing fresh and aged cirrus in one, one sees that fresh anvil cirrus is beyond the sensitivity (beta < 1.1) and the other data which show indeed a spread over beta are incredibly close to the fit. It would be more convincing to have this fit made from several campaigns. Especially aged anvils are not really the focus of your analyses.

Author response: Fig. 2 indicates that the $\beta$eff – N/IWC relationship is not sensitive to whatever differences there are between fresh and aged anvils, but there is a slight off-set for lower $\beta$eff between the TC4 and SPARTICUS data. This could be due to differences in PSD shape (Mitchell et al., 2011, ACP). In future work, we plan to examine this relationship using PSD measurements from other field campaigns such as ATTREX. The TC4 data we received from SPEC, Inc. contained the distinction of aged and fresh anvil cirrus, so we just included that; you are right that this is not a focus of our study.

6. There are a number of assumptions that go into the derivation, e.g. the shape of the ice particles (hexagonal columns down to sizes where they are not truly relevant).

Author response: Actually, we were rather conscientious about the distribution of ice particle shape with respect to size, guided by the observations reported in Baker and Lawson (2006, JAS) and Lawson et al. (2006, JAS) as stated on p. 7: "This shape-dependence on ice particle size was guided by the ice particle size-shape observations reported in Lawson et al. (2006) and Baker and Lawson (2006b)." Unfortunately, we

have no tunneling estimates for "irregulars", so we substituted small columns for small irregular crystal shapes. The CPI images for these small irregular crystals show quasi-spherical shapes that should support tunneling well (as do small columns); hence the substitution. Moreover, it is unclear what fraction of these small irregular particles are shattering artefacts since these studies were done in 2006 before the shattering problem was known and the probes were not designed to prevent shattering. The two studies cited above focus on two different types of cirrus clouds; wave cirrus and synoptic cirrus, and for each type, the distribution of particle shape with size is quite similar. This argues that the shape distribution is not so arbitrary. As mentioned in the last paragraph on p. 8, the constant $\beta$eff wrt temperature retrieved using MODIS channels (Fig. 4) was reproduced by the TC4 PSD for T > -55 °C provided this variation of crystal shape with size was implemented in the $\beta$eff calculations. Thus, the MODIS retrieved $\beta$eff-temperature relationship served as a check on the ice particle shape assumptions. The above sentence from the paper has been amended to read on p. 7, lines 28-30: "This shape-dependence on ice particle size was guided by the ice particle size-shape observations reported in Lawson et al. (2006) and Baker and Lawson (2006b), where small hexagonal columns (for which we can estimate Te) are substituted for small irregular crystals."

7. In polar locations, one has to be very careful with the underlying surface: the emissivity of ice, or sea ice (aging sea ice or fresh snow), has to be treated with special care. Was this taken into account for the retrieval?

Author response: The emissivity of the underlying surface is taken into account in the computation of the background radiance RBG (see response to comment 4b) at any locations. The surface emissivity at 10.6 and at 12.05 $\mu$m are inferred from IGBP surface types and a daily updated snow/ice index. More details are available in Garnier et al. (2012). Remaining errors in the surface emissivity at 10.6 and at 12.05 $\mu$m contribute to remaining biases between observed and computed TBG in clear sky conditions. These biases are corrected monthly with a resolution of 2 degrees in latitude and 4

degrees in longitude, and by separating nighttime and daytime. These characteristics were chosen as a compromise between space/time resolution and number of samples. The sentence describing RBG on p. 10, lines 12-15, now reads: "The brightness temperature TBG associated to the background radiance RBG is derived from the FASt RADiative transfer model (Dubuisson et al., 2005) fed by atmospheric profiles and skin temperatures from GMAO GEOS5 along with pre-defined surface emissivities inferred from the International Geosphere and Biosphere Program (IGBP) surface types and a daily updated snow/ice index (Garnier et al., 2012)."

8. There are many figures (22), many of them multi-panel figures (18 panels for each of the figures 15 to 18), adding up to more than 120 panels. This is too much. Surely it is the authors' responsibility to summarize and synthesize their work in a reasonnable number of figures.

Author response: We agree there are many figures, but we also feel that they contain important information. We tried to keep the figures to a minimum while preserving the critical information we feel "cirrus scientists" will find useful. The relatively large number of figures results from the fact that this is a new remote sensing method that has revealed some new findings potentially important to the scientific community.

9. There exist other retrievals of ice crystal number concentrations, using lidar - radar synergy, for example from DARDAR products (Delano and Hogan J. Geophys. Res. 2008 and 2010). How do your results compare to these ?

Author response: The Delano-Hogan DARDAR papers mentioned above are proof-of-concept type papers that do not report results similar to those in our paper. However, the DARDAR study on ice clouds by Hong and Liu (2015, J. Climate) and the CALIPSO lidar cirrus study by Nazaryan et al. (2008) do contain results relevant to our study. Comparisons and references to Hong and Liu appear in Sections 5.1 and to Hong and Liu (2015) and Nazaryan et al. (2008) in Sect. 7.1. The last two figures in our paper showing De results have been replaced to better relate these results to Fig. 6

in the Hong and Liu study. New text has been added to the end of Sect. 5.4, p. 24, lines 1-9: "Ice cloud CALIPSO-CloudSat retrievals of De are reported by Hong and Liu (2015) against temperature in terms of season and latitude zone for ODs ranging from < 0.03 to > 20. These results are presented for only two periods of the year; September-February and March-August, and by combining both hemispheres in terms of warm and cold season. Keeping in mind the different cloud sampling (e.g. OD range), it is thus not straight-forwards comparing their seasonal De changes with those in this study, although mean De values were comparable at temperatures colder than 235K (the upper limit in this study). In Fig. 6 of Hong and Liu (2015), effective radius (i.e. De/2) is plotted against altitude and latitude for each season, making it easier to compare between studies. Interestingly, De in the upper troposphere north of 30 ° N are larger in summer than in winter, which is consistent with findings from this study."

Minor points

p1, line 26: relevancy or relevance? Author response: relevance; done

p2, lines 6-7: a reference justifying this assertion would be useful Check Kramer & Luebke

p2, l14: Koop [et al], 2000 Author response: Done

p2, l22: There is a recent study on gravity wave fluctuations and their implications on nucleation processes which provides an update and complement to the Haag et al (2003) reference: Jensen, E. J., et al. (2016), High-frequency gravity waves and homogeneous ice nucleation in tropical tropopause layer cirrus, Geophys. Res. Lett., 43, 6629–6635, doi:10.1002/2016GL069426.

Author response: Thanks; done.

p3, l4-6: 'they draw from a limited number of field campaigns... of het and hom cirrus.' This sentence is too critical of past field campaigns. Of course field campaigns have their limitations, and of course it is necessary to complement them with global

**[ACPD](url)**

Interactive
comment

observations, but these should be presented more as complementary, more positively.

Author response: The text goes on to give examples, and few scientists would argue that field campaigns in the Polar Regions are sufficient. The ML Cirrus field campaign is planning to focus on Arctic cirrus in 2018. If cirrus have a seasonal dependence, then certainly field campaigns are currently not able to characterize this for all latitude zones.

p3, l16: I have not found the explanation in the text of the GMAO acronym. Author response: The end of the sentence on p. 3, line 16, is now: "from the Global Modeling Assimilation Office (GMAO) Goddard Earth Observing System Model, version 5 (GEOS 5)."

p9, section 3.2: I found the presentation a little awkward: the full equation is given first (equation (4)), then explanations for the different parts of the equation are provided. It is fine to stick with this, but I suggest below an alternative suggestion: - information on beta_eff allows to obtain N/IWC; hence to obtain N we wish to estimate IWC. This is given by (5) (a reference could be useful here, although this may seem standard). - now in this expression, it is needed to estimate the effective layer- mean extinction coefficient, and this is given in (6). - putting the pieces together, an expression for N can be obtained from CALIPSO IIR and CALIOP (for Delta z_{eq}): eq (4).

Author response: The last sentence on p. 9 was modified to read as follows: "Inspecting the right hand side of (4), one can see that, excepting the N/IWC term, the other terms must be an expression for the layer-mean IWC. This can be demonstrated by the familiar equation:"

p11, l14: the OD range limits the conclusions of this study to visible cirrus; hence by construction the subvisible thin cirrus near the tropopause tropical layer are not covered.

Author response: Yes, that is correct, as we stated in Sect. 4.3 of the ACPD paper, p.

14, lines 5-10.

p14, lines 26-27: why this choice of only two years, 2008 and 2013?

Author response: When the paper was being developed, the CALIPSO data coverage spanned the period 2007-2015. In case there were trends in cirrus microphysics over time, we wanted to inspect years from both ends of this time-spectrum, but also years that did not have periods of missing data. It was a major work effort to thoroughly analyze two years of data, so we limited our analysis to two years that met the above criteria. As mentioned, we spot-checked three other years to insure the results found for 2008 and 2013 were general.

p15, l 28: It is inappropriate to include a figure which does not contain a color bar allowing to read the altitudes corresponding to the colors. Moreover, the figure contains more information than needed as the oceans' bathymetry is included. Finally, the purpose of the figure is only to recall the general location of Earth's major topographic features, and I believe most readers will be familiar enough with this (a detailed knowledge of topography is not required in the present study).

Author response: Many may not be familiar with the topography of eastern Asia, such as eastern Russia where several high mountain ranges are located. We see a high incidence of "hom cirrus" over this region in winter. Also, many are not aware of the topography of Antarctica and Greenland, which is critical for understanding the high incidence of hom cirrus there. Overall, we feel it is necessary to have a figure like this in the paper to properly interpret the results.

p16, l22: it is a bit surprising that het cirrus would be the most abundant over oceans, where ice nuclei concentrations may be expected to be weaker than over land. This is rather the signature that the intensities of updrafts matter more than the abundance of ice nuclei, is that the proper way to understand it?

Author response: Yes, you are right. In Sect. 2 we noted that hom was most sensitive to the cooling rate (i.e. updraft speed), while het depends more on IN. So over mountainous regions, in situ cirrus N can be much higher than over oceans.

p17: Figures 15-18: There are 72 panels in those figues. This is too much. I think the authors should identify patterns and show a more limited number of figures to display the main patterns, and add only those panels which contribute to interesting deviations from the general pattern. The authors should For instance, for 60S-30S, the four figures (15 to 18) show the same pattern, with variations that can be summarized in the text, or by a choice of fewer figures (just JJA and DJF). It is true that, with the figures available, one could spot small variations in the details; but at this stage, given the uncertainties which one should keep in mind for a new, innovative method of remote-sensing, one should give too much weight to the details of the distributions.

Author response: The maps show the seasonally dependent column-integrated values but not the vertical distributions, whereas these figures show the corresponding vertical distributions. The two complement each other. Our group has had some inspired email conversations over these profile results, and they may contain things we have not yet noticed. Therefore, we prefer to keep them as they are for others to ponder over if that is permitted.

p17, l27-28: the statement seems a bit exagerated, pushing the evidence further than should be.

Author response: On p. 20, line 18, we have changed the word "indicating" to "suggesting". It is a fact that, based on our retrievals and the criteria used for identifying hom cirrus, the hom cirrus are closely associated with De < 45 microns.

p18, l1: the precision is unnecessary, 0.43 +/- 0.05 seems sufficient, given the uncertainties.. This is true in many other places of the text, e.g. p20, lines 2 and 6....

Author response: Requested changes have been made.

p18, l20: it is very good that this is recalled, this should be emphasized also in other

parts of the text, e.g. in the conclusion.

Author response: On p. 27, line 23, a sentence from Sect. 7 (conclusions) has been modified to read: "Although the retrieval is restricted to single-layer cirrus cloud optical depths between about 0.3 and 3.0 (which excludes most TTL cirrus), this optical depth range is likely to be the most radiatively significant range due to the lower cirrus cloud frequency of occurrence at higher OD and a much lower cirrus cloud mean emissivity at the lower ODs (Hong and Liu, 2015)."

p20, l9-10: clouds up to 25km: are these Polar Stratospheric Clouds? In which case they should be distinguished from cirrus. Their physics are not the same. It would not make sense to mix the two populations.

Author response: Polar stratospheric clouds (PSCs) may consist of 3 different particle types: nitric acid trihydrate (NAT) crystals, liquid super-cooled ternary solution (STS) $H_2SO_4$-$HNO_3$-$H_2O$ droplets, and ice ($H_2O$) crystals. The PSCs sampled by our retrieval are often contiguous with cirrus clouds and are ice PSDs as per the detection method. That is, the atmosphere over Antarctica does not exhibit a tropopause as it does elsewhere, with temperatures continuing to decrease into the stratosphere. This allows ice PSCs to extend up to 25 km, with the stratosphere beginning $\sim$ 195 K. While the CALIOP lidar detects these PSCs at these high altitudes (e.g. 20-25 km), our retrieval does not (as shown in Fig. 18), but it does sample these same cloud decks where they are optically thicker ($\sim$ 12-15 km as shown in Fig. 18). This retrieved information regarding $H_2O$ PSCs is valuable to those studying PSCs, and should be reported. However, the indicated text has been modified for clarity; on pp. 22-23, line 32 and lines 1-2, it now reads: "During winter and spring some ice clouds over Antarctica extended into the stratosphere (> 12 km) up to 25 km altitude (although their centroid altitude was much lower). These appear to be a type of Polar Stratospheric Cloud (PSC) that consists of ice crystals, and their elevated centroid altitudes are shown by the extended tail of the histograms in Figs. 18 and 19."

p20, Figues 19 and 20: the maps are so patchy! It would be better to average together 2008 and 2013, to analyze more data (why just two years?) and probably to average zonally.

Author response: A huge amount of work was required to process and analyze two years of this CALIPSO data. We show these years separately to show the reader that the observed seasonal patterns appear to be consistent year-to-year and to show annual variability as well. To average is to lose meaningful data. That patchiness is partly due to the fact that the analysis here is looking at a temperature interval, from 206-218 K (as opposed to all cirrus in a column of atmosphere).

p22, line 2: reverse order: 'exhibit a hom-het annual cycle similar to ...'

Author response: On p. 24-25, the sentence was slightly modified to read: "Land in this latitude zone appears to exhibit a hom-het annual cycle that is similar to North America but with a stronger hom signature in South America probably due to (1) mountain-induced waves from the Andes Mountains and (2) lower IN concentrations in the Southern Hemisphere."

p22, l25: I am not sure that 'implies' is the proper word here. Should this rather be 'This accounts for our understanding that in situ cirrus...'

Author response: On p. 25, lines 23-24, this sentence was slightly modified to read: "This is consistent with the understanding that in situ cirrus initially form without the presence of pre-existing ice, but rather result from a clearsky ISSR."

p23, l8-9: This sentence is a bit mysterious. In particular, what is meant by 'mechanistic'?

Author response: This mysterious sentence has been removed and replaced with the following sentence on p. 26, lines 6-7: "This illustrates how the dominant cirrus formation mechanism can vary with location."

p23, l22: any ideas for the reason of these differences?

Author response: Yes; earlier it was argued that, since anvil cirrus are liquid origin cirrus, and since these cirrus clouds contain pre-existing ice, then RHi will generally not reach the hom threshold in these anvil cirrus clouds. For the southern mid-latitudes, the retrieval apparently sees more het cirrus since there is mostly ocean there associated with weaker atmospheric waves.

p24, l20: change to a comma: 'wave resonance absorption, a process...'

Author response: Done.

p24, l20-22: this point will be more convincing if more campaigns are used to establish the relationship.

Author response: Agreed.

p24, l25-29: this is an interesting point.

p25, l15: split into 2 sentences, : ...solar zenith angle. With the SW...'

Author response: Done.

p26: these last paragraphs on Arctic Amplification are too long given the speculative nature of the arguments at this point.

Author response: We have received different feedback from other scientists who really appreciated this speculation, and a conference poster (that was well received) was based on this topic. We feel that scientific research becomes more productive when it is driven by working hypotheses.

Please also note the supplement to this comment:
http://www.atmos-chem-phys-discuss.net/acp-2016-1062/acp-2016-1062-AC1-supplement.pdf

[Figure]

**Supplement:**

[revised manuscript text omitted]
, 2007).  Given these criteria for het and hom, one can see how both het and hom can contribute to N in a strong cirrus cloud updraft, with het producing the first crystals, but due to insufficient ice surface area to prevent the RHi from climbing, the hom threshold can be reached with crystals produced by hom usually dominating N.

In theory, one could use either N or RHi to discriminate between hom and het.  But in practice, for satellite remote sensing to discriminate between hom and het conditions among cirrus clouds, the best distinguishing feature to exploit appears to be the generally observed differences in N.  In Barahona and Nenes (2009, Fig. 4), competition effects between het and hom are simulated in a parcel model using a broad spectrum of conditions (affecting nucleation) found in nature.  They find that het generally accounts for N < 200 L$^{-1}$ and that either hom or het can account for N between 200 L$^{-1}$ and 500 L$^{-1}$, while hom generally accounts for N > 500 L$^{-1}$.  Hence, in this study we use N > 500 L$^{-1}$ as a conservative threshold for cirrus dominated by hom.  This threshold is especially conservative when one considers that entrainment and ice sedimentation effects are predicted to dilute the N in cirrus formed through large-scale ascent (Spichtinger and Gierens, 2009; Jensen et al., 2012b).  
[revised manuscript text omitted]
 (Jensen et al., 2013a) but was used in this analysis since to exclude it would likely result in greater N error than the N uncertainty introduced. $\beta_{eff}$ was

calculated from these PSDs using the method described in Parol et al. (1991) and Mitchell et al. (2010). This method was tested in Garnier et al. (2013, Fig. 1b) where $\beta_{eff}$ calculated from a radiative transfer model (FASDOM; Dubuisson et al., 2005) was compared with $\beta_{eff}$ calculated analytically via Parol et al. (1991), with good agreement found between these two methods. More specifically, to calculate $\beta_{eff}$ from PSD in this study, the PSD absorption

5 efficiency $\bar{Q}_{abs}$ is given as $\bar{Q}_{abs} = \beta_{abs} / A_{PSD}$, where $\beta_{abs}$ is the PSD absorption coefficient (determined by MADA from measured PSD) and $A_{PSD}$ is the measured PSD projected area. The PSD effective diameter was determined from the measured PSD as described in Mishra et al. (2014), but in essence is given as $D_e = (3/2)\ IWC/(\rho_i\ A_{PSD})$, where $\rho_i$ is the density of bulk ice (0.917 g cm$^{-3}$). The PSD extinction efficiency $\bar{Q}_{ext}$ was determined in a manner analogous to $\bar{Q}_{abs}$. The single scattering albedo $\omega_o$ was calculated as $\omega_o = 1 - \bar{Q}_{abs} / \bar{Q}_{ext}$ and the PSD asymmetry

10 parameter g was obtained from $D_e$ using the parameterization of Yang et al. (2005). Knowing $\bar{Q}_{abs}$, $\omega_o$ and g, $\beta_{eff}$ was calculated from the PSD as:

$\beta_{eff} = Q_{abs,eff}(12\ \mu m)/Q_{abs,eff}(11\ \mu m)$ , $\hspace{4cm}$ (2)

$Q_{abs,eff} = Q_{abs}\ (1 - \omega_o\ g) / (1 - \omega_o)$. $\hspace{4cm}$ (3)

Note that $\beta$ (i.e. $\beta_{eff}$ without scattering effects) is also equal to $\bar{Q}_{abs}(12\ \mu m)/\ \bar{Q}_{abs}(10.6\ \mu m)$.

15 In order to retrieve N, a relationship between N/IWC and $\beta_{eff}$ is useful. Figure 2 shows measurements of N/IWC from SPARTICUS flights over the central United States (blue) where some of the cirrus sampled (i.e. the "ridge crest cirrus"; see Muhlbauer et al., 2014) had high N (500-2200 L$^{-1}$) for T < -60°C. Anvil cirrus data from SPARTICUS are also included in Fig. 2. Also shown are N/IWC measurements from the TC4 field campaign for maritime "fresh" anvil cirrus (during active deep convection where the anvil is linked to the convective column) and

20 for TC4 aged anvil cirrus (anvils detached from convective column). Figure 2 relates $\beta_{eff}$ to the N/IWC ratio, where $\beta_{eff}$ was calculated from the same PSD measurements used to calculate N and IWC, based on the MADA method. The PSD measurements include size-resolved estimates of ice particle mass concentration based on Baker and Lawson (2006a), size-resolved measurements of ice projected area concentration, and the size resolved number concentration. This PSD information is the input for the MADA method that yields the coefficients of absorption

25 and extinction. The tunneling efficiency $T_e$ used in MADA was estimated from Table 1 in Mitchell et al. (2006), where for 1 $\mu m$ < D < 30 $\mu m$, droxtals and hexagonal columns are assumed and $T_e = 0.90$; for 30 $\mu m$ < D < 100 $\mu m$, budding bullet rosettes and hexagonal columns are assumed and $T_e = 0.50$; for D > 100 $\mu m$ bullet rosettes and aggregates are assumed and $T_e = 0.15$. This shape-dependence on ice particle size was guided by the ice particle size-shape observations reported in Lawson et al. (2006) and Baker and Lawson (2006b), where small hexagonal

30 columns (for which we can estimate $T_e$) are substituted for small irregular crystals. These ice particle shape assumptions affect only $T_e$, and the cloud optical properties are primarily determined through the PSD

measurements noted above (i.e. not the value of $T_e$). Due to $\beta_{eff}$'s sensitivity to tunneling and small ice crystals, a tight and useful relationship is found between N/IWC and $\beta_{eff}$ for N/IWC > ~ $10^7 g^{-1}$. As far as we know, this relationship was not known previously. For $\beta_{eff} < 1.035$, $\beta_{eff}$ is not sensitive to N/IWC and N/IWC cannot be estimated from $\beta_{eff}$. Much of the data shown in Fig. 2 were derived from ice cloud PSD at temperatures between -20 and -38 °C where PSD tend to be relatively broad with low $\beta_{eff}$ values (< 1.06). But since this method targets cirrus clouds with cloud base temperature less than -38 °C, PSD at these temperatures tend to be relatively narrow and appropriate for sampling using this retrieval method. Moreover, for purposes of discriminating hom cirrus from het cirrus, the $\beta_{eff}$ vs. N/IWC relationship appears ideal since the hom-het transition generally occurs in the region where $\beta_{eff}$ is sensitive to changes in N/IWC (i.e. when N/IWC > $5\times10^7$ $g^{-1}$, based on in situ PSD and the hom-het transition region described in Sect. 2).

Using this same in situ data and methodology, $\beta_{eff}$ has also been related to $1/D_e$ as shown in Fig. 3. $D_e$ is defined as (3/2) IWC/($\rho_i$ $A_{PSD}$) where $\rho_i$ is the density of bulk ice (Mitchell, 2002). Accordingly, $D_e$ was calculated from the measured PSD (see Mishra et al., 2014). The relationship is only useful for $D_e < 90$ µm since $\beta_{eff}$ is only sensitive to the smaller ice particles. $\beta_{eff}$ is a measure of the relative concentration of small ice crystals in a PSD (Mitchell et al., 2010), and $A_{PSD}$ and $\beta_{eff}$ (PSD integrated quantities) may be associated with a substantial portion of larger ice particles (D > 50 µm) before $\beta_{eff}$ loses sensitivity to changes in $D_e$. Although it is feasible to retrieve $D_e$ using the IIR 8.7 µm channel (e.g. Dubuisson et al., 2008; Garnier et al., 2013), by using the same channels for both $D_e$ and N, the two retrievals are self-consistent with respect to PSD moments. In addition, the analytical PSD calculations for $\beta_{eff}$ are more accurate when using the 10.6/12 combination relative to the 8.7/12 combination (Figs. 1b and 2b in Garnier et al., 2013, JAMC) since there is less multiple scattering with the 10.6/12 combination. The use of these relationships (i.e. Figs. 2 and 3) in satellite remote sensing necessarily assumes that the in situ measurements they are based on are globally representative; an assumption that is difficult to test.

[revised manuscript text omitted]

**4.2 Retrieval uncertainties**

The uncertainty in N ($\Delta N$) results from the estimated uncertainties in $\beta_{eff}$ and $\alpha_{ext}$, and ultimately from the estimated uncertainties in $\tau_{abs}$(12 μm) and $\tau_{abs}$(10.6 μm) assuming a negligible error in $\Delta Z_{eq}$ (Eq.1 and 6). After re-writing Eq. (4) as a function of $\tau_{abs}$(12 μm) and $\tau_{abs}$(10.6 μm), the uncertainty $\Delta N$ is computed by propagating the errors in $\tau_{abs}$(12 μm) and $\tau_{abs}$(10.6 μm). These errors are themselves computed by propagating the errors in i) the measured brightness temperatures $T_m$ associated to $R_m$, ii) the blackbody brightness temperatures $T_{BB}$ associated to $R_{BB}$, and iii) the background brightness temperatures $T_{BG}$ (Garnier et al., 2015). The uncertainties in $T_m$ at 10.6 μm and 12.05 μm are random errors set to 0.3 K according to the IIR performance assessment established by the Centre National d'Etudes Spatiales (CNES) assuming no systematic bias in the calibration. They are assumed to be statistically independent. In contrast, the error in $T_{BB}$ is the same for both channels, because the same cloud temperature $T_{BB}$ is used to compute $\tau_{abs}$(12 μm) and $\tau_{abs}$(10.6 μm). A random error of +/-2K in $T_{BB}$ is estimated to include errors in the atmospheric model. After correction for systematic biases based on differences between observations and computations in cloud-free conditions (see Sect. 3.2), the error in $T_{BG}$ is considered a random error equal to the standard deviation of the corrected distributions of BTDoc. As a result, the uncertainty in $T_{BG}$ at 12.05 μm is set to ± 1K over ocean, and to ± 3K over land for both night and day. Standard deviations of the distributions of

   As seen from Eq. (4), (5) and (6), $\beta_{eff}$ and $\alpha_{ext}$ are the two key parameters retrieved from the CALIPSO IIR to derive N/IWC, IWC, and finally N. The interrelationship between $\beta_{eff}$, $\alpha_{ext}$, IWC, and N is illustrated in Fig. 6 (top row), which also shows the range encountered for these properties in the selected cloud population. The red dashed lines are where N = 200 $L^{-1}$, 500 $L^{-1}$ (the liberal and conservative hom thresholds), and 1000 $L^{-1}$. From this we see that both hom and het contribute to cirrus cloud formation. The pink dashed lines are where IWC = 0.5 mg $g^{-3}$, 5 mg $g^{-3}$, or 30 mg $g^{-3}$. Large values of N result from larger values of $\beta_{eff}$ (yielding smaller $D_e$ and much larger N/IWC) and sufficiently large values of $\alpha_{ext}$ so that IWC is sufficiently large for these small values of $D_e$. For our data selection, $\alpha_{ext}$ is mostly between 0.05 $km^{-1}$ and 5 $km^{-1}$. The horizontal red dotted lines for $\beta_{eff}$ < 1.035 indicate where the retrieval is not sensitive to N/IWC. For $\beta_{eff}$ <1.035, N/IWC is set to the maximum possible value (5.6 $10^7$ $g^{-1}$) so that N is a priori overestimated in these conditions. For $\beta_{eff}$ < 1, $D_e$ is set to $D_e$ = 122 μm, as denoted by the horizontal pink lines, and IWC is a priori underestimated for these conditions.¶

**Commented [A1]:** In Eq. 1, we use the notation $\tau_{abs}$(12.05 μm). Change to $\tau_{abs}$(12 μm) for consistency?

[BTDoc(10.6 μm) - BTDoc(12 μm)] are generally smaller than 0.5 K. After accounting for the contribution from the measurements, which is estimated to be √2x0.3 = 0.45 K, this indicates that the errors in $T_{BG}$ at 10.6 and at 12.05 mm are canceling out and can therefore be considered identical.  More details about the uncertainty analysis and the equations used to compute ΔN can be found in the appendix.

5    Figure 6 (bottom row) shows ΔN/N against $β_{eff}$ for the same samples as in Fig. 6 (top row).  ΔN/N decreases as $β_{eff}$ increases, reflecting that the technique is sensitive to small crystals.  ΔN/N is found most of the time < 50% for $β_{eff}$ > 1.15.  For a given value of $β_{eff}$, the variability of ΔN/N is due to the variability of $Δβ_{eff}/β_{eff}$ and of $Δα_{ext}/α_{ext}$.  ΔN/N is found to be larger over land because of a larger uncertainty in $T_{BG}$ and also because the radiative contrast is sometimes relatively weak.  $Δβ_{eff}/β_{
[revised manuscript text omitted]

15    and Krämer, 2013).

**4.3.2   Comparisons with SPARTICUS data**

As noted above, the Krämer et al. cirrus dataset did not quantify the contribution of ice particle shattering to N. The 2D-Stereo (i.e. 2D-S) probe was designed to minimize the problem of ice particle shattering (Lawson et al., 2006) and subsequent studies have shown that when 2D-S probe measurements are combined with post-processing of 2D-

20    S data using an ice particle arrival time algorithm (Lawson, 2011), the shattering artefacts appear to be minimized (e.g. Jensen et al., 2009; Lawson, 2011; Cotton et al., 2013). The Small PARTicles In CirrUS (SPARTICUS) field campaign, conducted from January through June of 2010 in the central United States (for domain size, see Fig. 2 in Mishra et al., 2014), was designed to better quantify the concentrations of small (D < 100 μm) ice crystals in cirrus clouds (Mace et al., 2009). Cirrus cloud PSD were measured using the 2D-S probe, which produces shadowgraph

25    images with true 10 μm pixel resolution at aircraft speeds up to 170 m s$^{-1}$, measuring ice particles between 10 and 1280 (or more) μm (Lawson et al., 2006). The 2D-S PSD data was post-processed using an ice particle arrival time algorithm that identifies and removes ice shattering events from the data stream.

Our CALIPSO retrievals are compared against SPARTICUS data for synoptic cirrus clouds in Fig. 9, using the synoptic cirrus data described in Mishra et al. (2014). The two April 28th SPARTICUS flights were added to this

30    dataset (giving a total of 15 flights) since April 28th was previously mislabeled as an anvil cirrus case study, but was actually a ridge-crest cirrus event (a type of synoptic cirrus) as described in Muhlbauer et al. (2014). The CALIPSO

retrievals were restricted to the SPARTICUS domain and the retrievals were obtained from January through April of 2010 since this period contained only synoptic cirrus based on the SPARTICUS flights. The SPARTICUS PSD measurements in Fig. 9 (left panel) are binned into 5 °C temperature (T) intervals and indicate the fraction of PSD samples within an interval whereby N exceeds either 250 $L^{-1}$ (blue + symbols) or 500 $L^{-1}$ (orange x symbols). These values are liberal and conservative thresholds for hom cirrus based on Fig. 4 of Barahona and Nenes (2009) and were selected to serve as a "reality check" for our hom cirrus criteria. As expected, for T > -40 °C, the fraction of samples for N > 500 $L^{-1}$ is zero, and the fraction for N > 250 $L^{-1}$ does not exceed 4%. This is consistent with the fact that only het produces ice crystals at these temperatures. The fraction of N > 200 $L^{-1}$ for T > -40 °C was considerable, indicating 250 $L^{-1}$ is about the lowest justifiable hom threshold for this dataset. The CALIPSO retrievals for these same N-fractions are shown by the blue (N > 250 $L^{-1}$) and orange (N > 500 $L^{-1}$) histograms, while the corresponding dashed lines indicate the uncertainties (i.e. ± one standard deviation). The red histogram is also for N > 500 $L^{-1}$, but it is based only on samples having $\beta_{eff}$ > 1.15 which have relatively low uncertainties regarding N as discussed in Sec. 4.2. This is evident from the red-dashed histograms. The right panel indicates the number of sampled PSDs within each 5 °C interval (x symbols) and the number of CALIPSO retrieval samples within these same 5 °C intervals.

As cloud temperatures decrease from -40 °C in Fig. 9, there is a general increase in both in situ N-fractions, indicating increasing contributions from hom with decreasing T. This same general behavior is found in the CALIPSO retrievals for both N-fractions. Taken as a whole, there is general consistency between the SPARTICUS and CALIPSO N-fractions. It may also be noteworthy that the retrieved N-fraction corresponding to lower N-uncertainty (i.e. the red histogram) exhibits somewhat better agreement with the SPARTICUS data at the warmest CALIPSO sampling temperatures. Cirrus sampled near -40 °C are necessarily thin geometrically since cloud base was required to be < -38 °C. In view of this, the slightly higher N-fractions in the temperature-bin near -40 °C appear consistent with the cirrus simulations of Spichtinger and Geirens (2009), where "sedimentation induced quenching of nucleation" (i.e. hom) would act to decrease N more dramatically in thicker cirrus clouds.

In general, the agreement between the retrieved and in situ measured quantities in Figs. 7, 8 and 9 is favorable despite the uncertainties involved. Moreover, it appears that retrieved and reliable in situ values of N-fractions corresponding to the same spatial domain and time period are fairly self-consistent. Relative differences in retrieved N, $D_e$ and IWC should therefore be meaningful, and from these relative differences, mechanistic inferences can be made and hypotheses explaining these inferences can be postulated.

**5   Retrieval results and discussion**

**5.1   Frequency of occurrence of selected cirrus samples**

[revised manuscript text omitted]

Over the southern oceans, $D_e$ appears smaller than $D_e$ over the tropics, especially during the winter season. This is consistent with the higher N fraction over the southern oceans, as noted above.

Temperature and altitude profiles of median $D_e$ for each latitude zone, with profiles for summer and winter for each zone, are shown for Northern Hemisphere ocean and land in Fig. 22, and similarly for the Southern Hemisphere in Fig. 23. At least 100 samples contributed to each data-point, and data from both years (2008 and 2013) were combined to generate these profiles. The profiles exhibit a considerable latitudinal and seasonal dependence, with latitudinal differences up to ~ 30 μm for a given temperature, or up to ~ 40 μm for a given altitude. Seasonal differences may be up to 20 μm for a given temperature or up to 30 μm for a given altitude. Combined with the latitudinal and seasonal dependence of cirrus cloud frequency of occurrence (e.g. Table 2), these $D_e$ differences are likely to produce substantial variations in cirrus cloud net radiative forcing relative to a constant $D_e$ profile assumption.

Ice cloud CALIPSO-CloudSat retrievals of $D_e$ are reported by Hong and Liu (2015) against temperature in terms of season and latitude zone for ODs ranging from < 0.03 to > 20. These results are presented for only two periods of the year; September-February and March-August, and by combining both hemispheres in terms of warm and cold season. Keeping in mind the different cloud sampling (e.g. OD range) it is thus not straight-forwards comparing their seasonal $D_e$ changes with those in this study, although mean $D_e$ values were comparable at temperatures colder than 235K (the upper limit in this study). In Fig. 6 of Hong and Liu (2015), effective radius (i.e. $D_e/2$) is plotted against altitude and latitude for each season, making it easier to compare between studies. Interestingly, $D_e$ in the upper troposphere north of 30 ° N are larger in summer than in winter, which is consistent with findings from this study.

[revised manuscript text omitted]

5   **Figure 9.  Left panel: SPARTICUS measurements from the Mishra et al. (2014) dataset for synoptic cirrus (sampled January-April 2010) are shown by the x and + symbols, and correspond to PSD contained within 5°C temperature intervals.  The blue + symbols indicate the fraction of PSD having number concentration N > 250 L$^{-1}$ while the orange x symbols indicate the fraction of PSD having N > 500 L$^{-1}$.  The solid histograms show CALIPSO retrievals obtained within the SPARTICUS domain during January-April 2010 for the same N fractions within the same 5°C temperature intervals,**

10   **with blue for N > 250 L$^{-1}$ and orange for N > 500 L$^{-1}$.  The red histogram is also for N > 500 L$^{-1}$, but it is based only on samples having $\beta_{eff}$ > 1.15 which have relatively low uncertainties regarding N (see Sec. 4.2).  The dashed histograms show ± 1 standard deviation from the mean value (i.e. solid histograms).  Right panel: The x symbols indicate the number of PSDs used in the analysis (multiplied by 10 for clarity of presentation) for each temperature interval, while the histogram indicates the number of CALIPSO samples used in each temperature interval.**

[Figure]

**Figure 10: Frequency of occurrence (indicated by legend in center) of sampled cirrus clouds for 2008 for each season.**

[Figure]

**Figure 11:  Same as Fig. 10 but for 2013.**

[Figure]

5    **Figure 12:   Fraction of sampled cirrus clouds (indicated by legend in center) having N > 500 L$^{-1}$ for 2008 for each season.**

[Figure]

5      **Figure 13:  Same as Fig. 12 except for 2013.**

[Figure]

**Figure 14: Left column: Fraction of sampled cirrus with N > 500 L$^{-1}$ for each 2-degree latitude point (comprised of at least 15 samples) with uncertainties (±ΔN) for land (red) and ocean (blue) for each season during 2013. Right column: same as left column, but for fraction of sampled cirrus having N > 500 L$^{-1}$ and β$_{eff}$ >1.15.**

[Figure]

**Figure 15: Elevation map for the Earth. Over land, green is lowest and greyish-brown is highest. Source: Wikipedia.**

[Figure]

**Figure 16: Histograms for the number of DJF 1-km$^2$ selected cirrus cloud samples (black), the number of samples having N > 500 L$^{-1}$ (orange), the number having N > 500 L$^{-1}$ and $\beta_{eff}$ > 1.15 (red), and the number having $\beta_{eff}$ > 1.15 (blue) as a function of cloud centroid altitude. The fraction of these last 3 quantities relative to the number of selected samples is given under "Fraction" by the corresponding color. This is done for 6 latitude zones (rows) and for ocean only (left column), land only (middle column) and ocean + land (right column), and averaged for 2008 and 2013.**

[Figure]

**Figure 17: Same as Fig. 16 but for MAM.**

[Figure]

**Figure 18: Same as Fig. 16 but for JJA.**

[Figure]

**Figure 19: Same as Fig. 16 but for SON.**

[Figure]

5      Figure 20:  Median value of retrieved effective diameter ($D_e$) for 2008 for each season over the temperature range of 206 K to 218 K.  The color bar gives $D_e$ in microns.

[Figure]

5     **Figure 21: Same as Fig. 20 except for 2013.**

[Figure]

[Figure]

Figure 22. Upper panels: Temperature profiles of zonal mean values over ocean (left panel) and land (right panel) of the retrieved median effective diameter $D_e$ for N. Hemisphere summer (solid profiles) and winter (dashed profiles) during 2008 and 2013. Lower panels: Altitude profiles of zonal mean values over ocean (left panel) and land (right panel) of the retrieved median $D_e$ for N. Hemisphere summer (solid profiles) and winter (dashed profiles) and during 2008 and 2013. Latitude zones are denoted by colors as defined in the panels.

Figure 22: Temperature dependence of the retrieved effective diameter ($D_e$ in microns) for three latitude zones in the Northern Hemisphere over both ocean and land during the winter of 2013-2014.¶

————————————Page Break————————————

¶
¶                                                                    ...

[Figure]

**Figure 23. Same as in Fig. 22 but for the Southern Hemisphere. Now the profiles are solid for winter and dashed for summer.**